# BandPO: Bridging Trust Regions and Ratio Clipping via Probability-Aware Bounds for LLM Reinforcement Learning

Yuan Li [* 1 2]   Bo Wang [* 1]   Yufei Gao [1]   Yuqian Yao [1 2]   Xinyuan Wang [1]
Zhangyue Yin [† 1]   Xipeng Qiu [† 1 2]

## Abstract

Proximal constraints are fundamental to the stability of the Large Language Model reinforcement learning. While the canonical clipping mechanism in PPO serves as an efficient surrogate for trust regions, we identify a critical bottleneck: fixed bounds strictly constrain the upward update margin of low-probability actions, disproportionately suppressing high-advantage tail strategies and inducing rapid entropy collapse. To address this, we introduce Band-constrained Policy Optimization (BandPO). BandPO replaces canonical clipping with Band, a unified theoretical operator that projects trust regions defined by $f$-divergences into dynamic, probability-aware clipping intervals. Theoretical analysis confirms that Band effectively resolves this exploration bottleneck. We formulate this mapping as a convex optimization problem, guaranteeing a globally optimal numerical solution while deriving closed-form solutions for specific divergences. Extensive experiments across diverse models and datasets demonstrate that BandPO consistently outperforms canonical clipping and Clip-Higher, while robustly mitigating entropy collapse. Code is publicly available at https://github.com/OpenMOSS/BandPO.

## 1. Introduction

Reinforcement Learning from Human Feedback (RLHF) has established itself as the dominant paradigm for the post-training of Large Language Models (LLMs), wherein the

*Equal contribution †Corresponding authors. [1]College of Computer Science and Artificial Intelligence, Fudan University, Shanghai, China [2]Shanghai Innovation Institute, Shanghai, China. Correspondence to: Yuan Li <liyuan24@m.fudan.edu.cn>, Zhangyue Yin <yinzy21@m.fudan.edu.cn>, Xipeng Qiu <xpqiu@fudan.edu.cn>.

*Proceedings of the 43rd International Conference on Machine Learning*, Seoul, South Korea. PMLR 306, 2026. Copyright 2026 by the author(s).

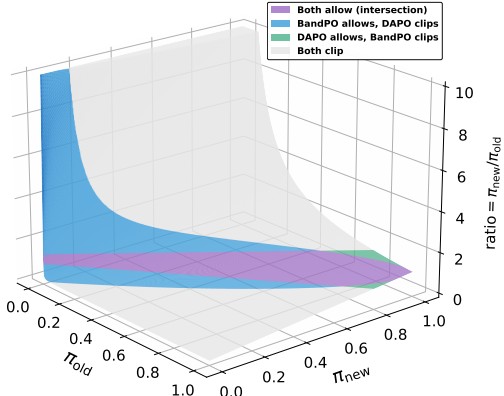

*Figure 1.* **Comparison of ratio clipping regions: BandPO vs. DAPO.** While DAPO enforces fixed asymmetric bounds ($\epsilon_+ = 0.28, \epsilon_- = 0.2$), BandPO projects a KL-induced trust region ($\delta = 0.1$) into dynamic bounds. The blue region highlights the expanded margin for low-probability, positive-advantage actions, effectively preventing premature saturation and preserving critical exploration gradients.

proximal constraint on policy updates serves as a pivotal mechanism designed to balance optimization stability with effective exploration (Ouyang et al., 2022). Schulman et al. (2017) emulate trust-region updates via clipping the surrogate objective, circumventing the expensive Fisher Information computations required by Trust Region Policy Optimization (TRPO) (Schulman et al., 2015). This fixed clipping mechanism has emerged as the default configuration for LLM RL (Zheng et al., 2023) and is extensively adopted in Group Relative Policy Optimization (GRPO) (Shao et al., 2024) and its variants.

While the canonical clipping mechanism ensures stability by emulating trust-region updates, Park et al. (2025) argue that it implicitly inhibits exploration by imposing a detrimental bias against policy entropy. More precisely, in Reinforcement Learning with Verifiable Rewards (RLVR) scenarios, while lower-bound clipping tends to increase entropy, upper-bound clipping decreases it; with symmetric clipping thresholds, the latter effect dominates, resulting in net entropy reduction even when the algorithm is fed purely random rewards (Shao et al., 2025). This bias effectively silences the gradient signals for low-probability yet high-advantage actions, preventing the model from re-

inforcing novel, superior strategies that lie in the tail of the distribution. To mitigate this, Yu et al. (2025) proposes the Clip-Higher strategy to relax the upper clipping bound. While Cui et al. (2025) acknowledges that it delays entropy collapse, they also highlight its instability—often leading to performance collapse after saturation. This indicates that adjusting thresholds fails to address the inherent limitations of fixed clipping bounds.

We formally characterize a critical structural bottleneck inherent in the canonical clipping mechanism, as detailed in Section 4. Specifically, constraining the probability ratio within fixed bounds enforces a linear dependence, wherein the maximum feasible probability variation scales proportionally with the old probability. Consequently, for positive-advantage actions, lower probabilities dictate vanishingly small margins for upward variation, rendering them susceptible to premature clipping and nullifying their gradient contributions. Constrained by the necessity of proximal policy updates, the margin for expanding the fixed range is limited, failing to reconcile the fundamental tension between optimization constraints and effective exploration.

To address this bottleneck, we propose Band-**constrained Policy Optimization (BandPO)**. We introduce a unified theoretical operator, Band, which projects trust regions induced by general $f$-divergences into dynamic, probability-aware clipping intervals. BandPO substitutes the clipping operator in GRPO with Band, employing a single interpretable radius parameter to enforce proximal constraints, thereby significantly streamlining hyperparameter tuning. Mathematically, we frame this mapping as a convex optimization problem, guaranteeing globally optimal numerical solutions while deriving efficient closed-form solutions for specific instances like Total Variation (TV) and Pearson $\chi^2$-divergence. Crucially, we theoretically analyze the properties of Band and confirm that it naturally circumvents this bottleneck. As exemplified by the KL-divergence case in Figure 1, the projected bounds adaptively expand the feasible upward margin for low-probability actions to prevent premature clipping, corresponding to the expanded blue region. Empirically, compared to standard GRPO and its variants that substitute the canonical clipping with Clip-Higher (Yu et al., 2025) and Dynamic-Adaptive Clipping (DAC) (Yang et al., 2025) respectively, GRPO with Band (BandPO) demonstrates consistent performance improvements across Qwen2.5 (3B, 7B) and Llama 3 (8B) on multiple mathematical benchmarks, while robustly mitigating entropy collapse.

Our main contributions are summarized as follows:

- We formally characterize a critical bottleneck in canonical clipping, revealing that the feasible upward update margin scales linearly with the action probability. This tends to nullify gradients for low-probability, positive-advantage

actions, inhibiting effective exploration.
- We propose BandPO, introducing a unified Band operator to project $f$-divergence-induced trust regions into dynamic clipping intervals. We formulate this as a convex optimization problem, guaranteeing globally optimal numerical solutions while deriving closed-form solutions for specific divergences, theoretically demonstrating that BandPO circumvents the bottleneck.
- We demonstrate that BandPO achieves consistent performance gains over GRPO and its variant baselines (Clip-Higher and DAC) across various models on math benchmarks, robustly mitigating entropy collapse.

## 2. Related Work

**From Trust Regions to Ratio Clipping.** Proximal constraints are fundamental to stable policy optimization, ensuring that the updated policy remains within a controlled vicinity of the sampling policy. Introduced by Schulman et al. (2015), the trust-region concept has been widely adopted in policy gradient optimization to impose proximal constraints. Schulman et al. (2015) employs the KL-induced trust region to ensure that the new policy remains within a small neighborhood of the old policy. Theoretically, such trust regions can be characterized by various distributional discrepancies, including integral probability metrics (Terpin et al., 2022) and the family of $f$-divergences—such as Pearson $\chi^2$ and TV (Csiszár, 1967; Nowozin et al., 2016). However, the resulting constrained optimization problem involves an inequality constraint, rendering it computationally prohibitive for large-scale applications. To address this issue, Schulman et al. (2017) proposed two variants—PPO-Penalty and PPO-Clip—that eliminate the need for conjugate gradient methods. The PPO-Clip mechanism and its variants have been extensively adopted across diverse domains, spanning complex strategic gaming (OpenAI et al., 2019; Baker et al., 2020), physics-based character animation (Peng et al., 2018), and robotic control (OpenAI et al., 2020; Yu et al., 2022). Subsequent works have established the clipping mechanism as the dominant paradigm for LLM post-training (Zheng et al., 2023). Concurrently, there is a notable shift towards critic-free paradigms to enhance computational efficiency (Li et al., 2024; Ahmadian et al., 2024; Hu et al., 2025; Shao et al., 2024).

**Adaptive Clipping Variants.** The clipping mechanism is favored for its simplicity, yet it also raises ongoing questions regarding the appropriate choice of clipping bounds. To mitigate instability from negative-advantage actions, Ye et al. (2020) introduces an auxiliary lower bound, adopted in the LLM RL framework (Sheng et al., 2024). Chen et al. (2018) proposes a state-wise adaptive clipping mechanism that modulates the clipping strength according to state importance, while Farsang & Szegletes (2021) applies a simple

time-decaying schedule to the clipping range. Despite their empirical success, these heuristics rely on auxiliary hyper-parameters lacking a clear theoretical grounding, rendering them brittle and difficult to tune. Wang et al. (2019) establishes a theoretical connection between KL-divergence and the clipping bounds, demonstrating performance improvements in continuous control tasks.

**Clip Control in LLM.** LLM RL operates in an extremely high-dimensional action space, where the combination of long reasoning horizons and extensive group sampling results in a high cumulative incidence of low-probability actions. To address this issue, Decoupled Clip and Dynamic sAmpling Policy Optimization (DAPO) (Yu et al., 2025) proposes the Clip-Higher strategy, which decouples and relaxes the clipping upper bounds. Building upon the decoupled bounds, Dynamic Clipping Policy Optimization (DCPO) (Yang et al., 2025) introduces DAC, deriving dynamic-adaptive clipping bounds via inequality relaxation that dynamically adjust according to the probabilities. Recent work has also explored adaptive clipping for off-policy LLM RL. BAPO (Xi et al., 2025) adaptively adjusts clipping bounds to preserve entropy and stabilize training.

**Direct Entropy Regularization.** Beyond clipping-based control, direct entropy regularization (DER) provides another way to regulate the exploration–exploitation trade-off in LLM RL by explicitly biasing the policy objective toward higher entropy. Recent analyses identify rapid entropy collapse as a key failure mode in reasoning RL (Cui et al., 2025), and show that high-entropy minority tokens or low-probability exploratory tokens can play a disproportionate role in discovering effective reasoning paths (Wang et al., 2025b; Huang et al., 2025). Motivated by this, several methods introduce explicit entropy-aware interventions, such as selective entropy regularization (Jiang et al., 2025), adaptive entropy coefficients (Zhang et al., 2025), temperature-guided entropy control (Wang et al., 2025a), and entropy-ratio clipping (Su et al., 2025). DER-style methods and BandPO operate at different levels: the former directly reshapes the optimization objective or policy entropy, while BandPO keeps the RL objective unchanged and adjusts the clipping geometry of each policy update.

However, compared to the maturity of continuous control, the theoretical underpinnings of clipping mechanisms in LLM RL remain under-explored. Existing methods lack a principled framework to control clipping bounds via simple, effective, and interpretable parameters, consequently struggling to balance proximal constraints with effective exploration. To bridge this gap, we propose BandPO, which employs a unified operator that projects $f$-divergences-induced trust regions into probability-aware clipping intervals, resolving the bottleneck using an interpretable parameter.

## 3. Preliminaries

### 3.1. Notation

We formulate the RL alignment of LLMs as a discrete Markov Decision Process. Let $\pi_\theta$ denote the policy represented by the LLM. Given a prompt $x$ sampled from a dataset $\mathcal{D}$, the policy generates a response sequence $y = (a_1, a_2, \ldots, a_T)$ by auto-regressively sampling from a vocabulary $\mathcal{V}$ of size $V = |\mathcal{V}|$. Each action $a_t \in \mathcal{V}$ corresponds to a token generated by the LLM. At step $t$, the state $s_t = (x, y_{<t})$ comprises the prompt $x$ and the preceding tokens $y_{<t} = (a_1, \ldots, a_{t-1})$. The policy $\pi_\theta$ maps $s_t$ to a conditional probability distribution $\pi_\theta(\cdot \mid s_t)$ over $\mathcal{V}$. We denote by $\mathbb{R}$ and $\mathbb{R}_+$ the sets of real and non-negative real numbers, respectively. Let $\Delta^V \triangleq \{p \in \mathbb{R}_+^V \mid \sum_{i=1}^V p_i = 1\}$ denote the probability simplex over $V$ categories. The optimization objective is to maximize the expected reward: $J(\theta) = \mathbb{E}_{x \sim \mathcal{D}, y \sim \pi_\theta}[R(x, y)]$, where $R(x, y)$ denotes a sparse, sequence-level scalar reward, typically derived from verification signals from verification signals.

### 3.2. Clip-Based Proximal Constraints in LLM RL

Consider an iterative optimization process where we update $\pi_\theta$ using trajectories sampled by $\pi_{\text{old}}$. The probability ratio $r_t(\theta) = \frac{\pi_\theta(a_t|s_t)}{\pi_{\text{old}}(a_t|s_t)}$ serves as an importance sampling weight to correct for the distributional shift. Schulman et al. (2017) proposed clipping $r_t$ to impose proximal constraints, while employing a critic model for Generalized Advantage Estimation (Schulman et al., 2018). Inheriting the clipping mechanism, Shao et al. (2024) introduced Group Relative Policy Optimization (GRPO), which circumvents the computational burden of training a critic model by estimating advantages from a group of sampled responses. We denote the objective as $\mathcal{J}^{\text{GRPO}}(\theta)$, which aggregates the per-token objectives across a group of $G$ outputs:

$$\mathbb{E}_{x \sim \mathcal{D}, \{y_i\}_{i=1}^G \sim \pi_{\text{old}}} \left[ \frac{1}{G} \sum_{i=1}^G \frac{1}{T_i} \sum_{t=1}^{T_i} \mathcal{J}_t(\theta; y_i) \right], \quad (1)$$

Specifically, we formulate $\mathcal{J}_t(\theta; y_i)$ to admit asymmetric clipping boundaries:

$$\mathcal{J}_t(\theta; y_i) = \min \left( r_{t,i} A_{t,i}, \text{clip}(r_{t,i}, 1 - \epsilon_-, 1 + \epsilon_+) A_{t,i} \right) - \beta \underbrace{D_{\text{KL}}(\pi_\theta(\cdot|s_t) \| \pi_{\text{ref}}(\cdot|s_t))}_{\text{per-token KL divergence}}. \quad (2)$$

This formulation averages the objective over tokens to accommodate varying sequence lengths $T_i$. $G$ denotes the group size, and $r_{t,i}$ represents the probability ratio. The advantage $A_{t,i}$ is derived from the sequence-level rewards $R_i$, standardized within the group via $A_{t,i} = (R_i - \mu_R)/\sigma_R$,

where $\mu_R$ and $\sigma_R$ are the mean and standard deviation of rewards $\{R_j\}_{j=1}^G$, respectively. The term $\beta D_{\mathrm{KL}}$ serves as a regularization penalty, anchoring $\pi_\theta$ to the reference policy $\pi_{\mathrm{ref}}$ to preserve linguistic coherence. While GRPO employs symmetric bounds ($\epsilon_+ = \epsilon_-$), Equation (2) generalizes this to allow asymmetric bounds ($\epsilon_+ > \epsilon_-$).

## 4. The Bottleneck in Canonical Clipping

The canonical clipping mechanism confines the probability ratio $r_t(\theta)$ to the interval $[1-\epsilon_-, 1+\epsilon_+]$, where $\epsilon_-, \epsilon_+ > 0$ are fixed constants. This constraint is formulated as:

$$1 - \epsilon_- \le \frac{\pi_\theta(a \mid s)}{\pi_{\mathrm{old}}(a \mid s)} \le 1 + \epsilon_+. \tag{3}$$

We define the probability variation of an action $a$ given state $s$ as: $\Delta\pi(a|s) = \pi_\theta(a|s) - \pi_{\mathrm{old}}(a|s)$, which explicitly quantifies the deviation of the probability relative to the sampling policy. Multiplying all terms in Inequality (3) by the strictly positive $\pi_{\mathrm{old}}(a|s)$ and subtracting $\pi_{\mathrm{old}}(a|s)$ yields the set of feasible probability variations, denoted as $\mathcal{C}_{\mathrm{clip}}$:

$$\{\Delta\pi(a|s) \mid -\epsilon_-\pi_{\mathrm{old}}(a|s) \le \Delta\pi(a|s) \le \epsilon_+\pi_{\mathrm{old}}(a|s)\}. \tag{4}$$

Theoretically, the constraint $\pi_\theta(\cdot|s) \in \Delta^V$ allows the variation to range from $-\pi_{\mathrm{old}}$ (dropping to 0) to $1 - \pi_{\mathrm{old}}$ (rising to 1). We define this maximal feasible set as $\mathcal{C}_{\mathrm{simplex}}$:

$$\{\Delta\pi(a|s) \mid -\pi_{\mathrm{old}}(a|s) \le \Delta\pi(a|s) \le 1 - \pi_{\mathrm{old}}(a|s)\}. \tag{5}$$

The derivation above establishes the relationship $\Delta\pi(a|s) = (r_t(\theta) - 1)\pi_{\mathrm{old}}(a|s)$. Using this relationship to map $\mathcal{C}_{\mathrm{simplex}}$ into the ratio space yields the theoretical bounds of $r_t(\theta)$:

$$0 \le \frac{\pi_\theta(a \mid s)}{\pi_{\mathrm{old}}(a \mid s)} \le \frac{1}{\pi_{\mathrm{old}}(a \mid s)}. \tag{6}$$

As illustrated in Fig. 2, analysis within $\mathcal{C}_{\mathrm{simplex}}$ reveals that fixed clipping bounds on $r_t(\theta)$ (e.g., DAPO) constrain probability variations to scale linearly with $\pi_{\mathrm{old}}(a|s)$. Consequently, the feasible upward shift vanishes as the probability approaches zero, contradicting the theoretical upper bound. This discrepancy induces premature clipping for low-probability actions with positive advantages, effectively nullifying gradient contributions. Conversely, in high-probability regimes, the upper bounds of DAPO and DCPO exceed inherent simplex limits, rendering the constraint mathematically vacuous. Thus, fixed bounds fail to reconcile proximal constraints with exploration efficacy, while dynamic bounds lacking clear theoretical grounding result in partial constraint failure. These limitations necessitate a transition to theoretically grounded, dynamic bounds.

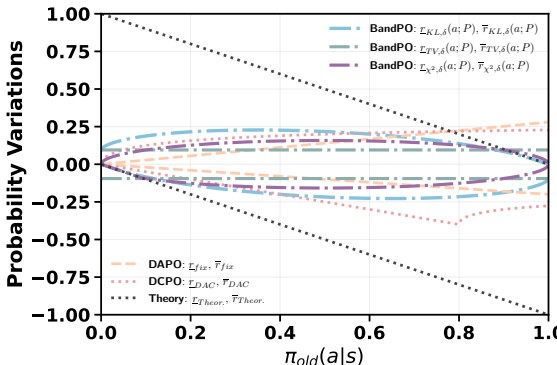

*Figure 2.* **Comparison of the Bounds of Probability Variation.** We visualize the bounds of variation derived from the Theoretical Simplex, DAPO, DCPO, and BandPO (ours). The symbols $\bar{r}$ and $\underline{r}$ denote the upper and lower clipping boundaries, respectively. Parameters are fixed at $\epsilon_+ = 0.28, \epsilon_- = 0.2$, and $\delta = 0.1$. BandPO strictly adheres to physical simplex constraints while unlocking significant upward variation for low-probability actions.

## 5. Method

In this section, we introduce **BandPO**. Central to BandPO is the $\mathsf{Band}$ operator, which projects trust regions defined by $f$-divergences into probability-aware clipping intervals. Our analysis of $\mathsf{Band}$'s theoretical properties demonstrates that it effectively resolves the exploration bottleneck in Section 4.

### 5.1. $f$-Divergence-Induced Trust Regions

Consider a fixed state $s_t$. Let $P(\cdot) \triangleq \pi_{\mathrm{old}}(\cdot|s_t) \in \Delta^V$ and $Q(\cdot) \triangleq \pi_\theta(\cdot|s_t) \in \Delta^V$ denote the distributions over $\mathcal{V}$, respectively. We formalize the trust region using the family of $f$-divergences (Csiszár, 1967). Let $f : \mathbb{R}_+ \to \mathbb{R}$ be a strictly convex function with $f(1) = 0$. The divergence $D_f(Q\|P)$ is defined as:

$$D_f(Q\|P) \triangleq \sum_{a \in \mathcal{V}} P(a)f\left(\frac{Q(a)}{P(a)}\right). \tag{7}$$

With $P$ serving as the anchor, we define the trust region $\mathcal{T}_{f,\delta}(P)$ as the convex set constraining $Q$ within a $\delta$-neighborhood of $P$ on the probability simplex $\Delta^V$:

$$\mathcal{T}_{f,\delta}(P) \triangleq \left\{Q \in \Delta^V \mid D_f(Q\|P) \le \delta\right\}, \tag{8}$$

where $\delta > 0$ dictates the radius of the trust region. This geometric formulation generalizes standard proximal constraints. Notably, choosing the generator function $f(u) = -\log u + u - 1$ recovers the specific trust region employed in TRPO (Schulman et al., 2015), i.e., $D_{\mathrm{KL}}(P\|Q) \le \delta$.

## 5.2. Band: An Operator for Projecting Trust Regions to Probability-Aware Clipping Bounds

We now derive the probability-aware ratio bounds induced by the geometric constraint $\mathcal{T}_{f,\delta}(P)$. To facilitate the analysis, we recast the probability ratio $r_t(\theta)$ as a function of the candidate distribution $Q \in \Delta^V$. For any token $a \in \mathcal{V}$ satisfying $P(a) > 0$, the ratio function is redefined as:

$$r(a; Q, P) \triangleq \frac{Q(a)}{P(a)}. \tag{9}$$

While $P$ remains fixed, the candidate distribution $Q \in \Delta^V$ varies subject to the constraint $\mathcal{T}_{f,\delta}(P)$.

**Optimal Dynamic Bounds.** The upper bound of the probability ratio is determined by maximizing the probability $Q(a)$ subject to the constraint $\mathcal{T}_{f,\delta}(P)$. Since $\mathcal{T}_{f,\delta}(P)$ forms a convex feasible set, this formulation constitutes a convex optimization problem with a linear objective function with respect to the decision variable $Q$:

$$\bar{r}_{f,\delta}(a; P) \triangleq \max_{Q \in \mathcal{T}_{f,\delta}(P)} \frac{Q(a)}{P(a)}. \tag{10}$$

Symmetrically, the minimal admissible probability ratio is determined by solving:

$$\underline{r}_{f,\delta}(a; P) \triangleq \min_{Q \in \mathcal{T}_{f,\delta}(P)} \frac{Q(a)}{P(a)}. \tag{11}$$

Crucially, Problems (10) and (11) are convex programs. Under strictly convex $f$ and $P(a) \in (0, 1)$, any local optimum is the global optimum, enabling stable numerical solutions.

**The Band Operator.** We propose Band, a unified operator designed to supersede canonical clipping by strictly constraining the probability ratio within the feasible interval induced by the $f$-divergence trust region. Given a generator function $f$ and radius $\delta$, for $\forall P(a) \in (0, 1)$, solving Problems (10) and (11) yields the rigorous lower and upper bounds for the ratio. Crucially, this derivation projects the high-dimensional geometric constraint $\mathcal{T}_{f,\delta}(P)$ onto a scalar interval specific to action $a$, to which the ratio is strictly confined. We formulate this operation as:

$$\text{Band}_{f,\delta}(r; a, P) \triangleq \text{clip}\left(r, \underline{r}_{f,\delta}(a; P), \bar{r}_{f,\delta}(a; P)\right). \tag{12}$$

In stark contrast to the fixed clipping interval $[1 - \epsilon, 1 + \epsilon]$, Band yields probability-aware bounds governed solely by the single, interpretable trust-region radius $\delta$.

## 5.3. Reduction to Univariate Optimization

Although Problems (10) and (11) are convex, the decision variable $Q$ resides in a high-dimensional simplex ($V \approx 10^5$), rendering direct optimization computationally prohibitive. We circumvent this by exploiting the structural symmetry of the divergence constraint to strictly reduce the problem to a univariate formulation, as established in the following lemma, proven in Appendix A.1.

**Lemma 1** (Optimality of Uniform Complement Rescaling). *Given a reference distribution $P \in \Delta^V$ with full support (i.e., $P(v) > 0, \forall v$) and an action $a \in \mathcal{V}$, there exists an optimal solution $Q^\star$ to the extremal Problems (10) and (11) that preserves the relative probability proportions within the complement set $\mathcal{V} \setminus \{a\}$. Specifically, the probability ratio is constant for all non-target actions:*

$$\frac{Q^\star(b)}{P(b)} = c, \quad \forall b \in \mathcal{V} \setminus \{a\}, \tag{13}$$

*where the scaling factor $c \in \mathbb{R}_+$ is uniquely determined by the simplex normalization constraint $\sum_{v \in \mathcal{V}} Q^\star(v) = 1$. This implies that $Q^\star$ is fully parameterized by the single scalar ratio $r = Q^\star(a)/P(a)$ at the target action. When the generator is strictly convex on the complement ratios, this complement rescaling is unique.*

Based on Lemma 1, the optimization reduces to determining the scalar ratio $r \triangleq Q(a)/P(a)$ in the target action $a$. Let $p \triangleq P(a)$. The simplex constraint $\sum_v Q(v) = 1$ uniquely determines the complement scaling factor $c$ as a function of $r$, since $rp + c(1 - p) = 1 \implies c(r) = \frac{1 - rp}{1 - p}$. Substituting this mapping into Equation (7), the divergence collapses into a univariate function $g_f(p, r)$:

$$D_f(Q\|P) = \underbrace{P(a)f(r)}_{\text{target}} + \sum_{b \neq a} P(b)f(c)$$
$$= pf(r) + (1 - p)f\left(\frac{1 - rp}{1 - p}\right) \triangleq g_f(p, r). \tag{14}$$

Equation (14) reduces $D_f(Q\|P) \leq \delta$ to the scalar constraint $g_f(p, r) \leq \delta$ over $r \in [0, 1/p]$. Thus, Problems (10) and (11) amount to finding the endpoints of a scalar feasible interval: active endpoints are roots of $g_f(p, r) = \delta$, while inactive endpoints saturate at simplex boundaries. The result is formalized in Theorem 1, with proof in Appendix A.2.

**Theorem 1** (Exact Scalarization of Trust-Region Constraints). *Assume the generator function $f : \mathbb{R}_+ \to \mathbb{R} \cup \{+\infty\}$ is strictly convex with $f(1) = 0$ and twice differentiable on $\mathbb{R}_{++}$. For any action $a$ with $P(a) = p \in (0, 1)$, Problems (10) and (11) reduce to optimizing the scalar ratio $r = Q(a)/P(a)$ over $r \in [0, 1/p]$ under*

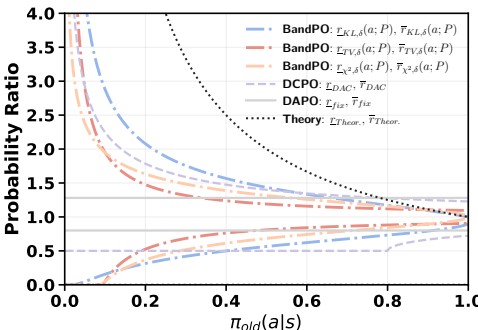

*Figure 3.* **Comparison of Probability Ratio Bounds.** We visualize the ratio bounds derived from the Theoretical Simplex, DAPO, DCPO, and BandPO (ours). As $p \to 1$, BandPO bounds exhibit monotone head behavior: upper bounds decrease to 1, while lower bounds increase toward divergence-dependent limits. Conversely, as $p \to 0$, the upper bounds of both DCPO and BandPO expand rapidly, preventing premature clipping. Note that for TV and $\chi^2$, the radius $\delta = 0.1$ triggers the simplex saturation condition.

$$g_f(p, r) \triangleq pf(r) + (1-p)f\left(\frac{1-rp}{1-p}\right) \leq \delta. \quad (15)$$

*Moreover, $g_f(p, r)$ is strictly convex in $r$, achieves its minimum $0$ at $r = 1$, and therefore induces a scalar feasible interval containing $r = 1$. The optimal clipping bounds are its endpoints:*

$$\underline{r}_{f,\delta}(a; P) = \min\{r \in [0,1] \mid g_f(p, r) \leq \delta\}, \quad (16)$$

$$\overline{r}_{f,\delta}(a; P) = \max\{r \in [1, 1/p] \mid g_f(p, r) \leq \delta\}. \quad (17)$$

*In the active non-saturated regime, these endpoints are the unique roots of $g_f(p, r) = \delta$ on the corresponding branches; otherwise, they saturate at the simplex boundaries.*

For non-strictly convex divergences such as TV, Theorem 1 does not rely on strict convexity for the scalarization itself; the same scalarized feasible-interval characterization still holds. The branch-wise root structure and monotonicity for TV are verified directly from the closed-form expression in Appendix A.7.

### 5.4. Properties of Band Bounds

Building upon Theorem 1, we theoretically analyze the asymptotic behavior, active-regime strict monotonicity, and saturation-aware monotonicity of the bounds $\overline{r}_{f,\delta}(p)$ and $\underline{r}_{f,\delta}(p)$ with respect to $p \in (0, 1)$, with proofs provided in Appendix A.3 and A.4.

This analysis demonstrates that Band theoretically resolves the exploration bottlenecks identified in Section 4.

**Proposition 1** (Asymptotic Behavior of Band Bounds). *Given a trust-region radius $\delta > 0$, the universal limiting behaviors of the bounds $\overline{r}_{f,\delta}(p)$ and $\underline{r}_{f,\delta}(p)$ as $p \in (0,1)$ approaches the simplex boundaries are:*

$$\lim_{p \to 0^+} \overline{r}_{f,\delta}(p) = +\infty, \quad \lim_{p \to 0^+} \underline{r}_{f,\delta}(p) = 0, \quad \lim_{p \to 1^-} \overline{r}_{f,\delta}(p) = 1. \quad (18)$$

**Proposition 2** (Strict Monotonicity of Band Bounds). *In the active non-saturated regime, the clipping bounds are strictly monotonic functions of the probability $p$. Specifically, given a $\delta > 0$, the upper bound $\overline{r}_{f,\delta}(p)$ is strictly decreasing with respect to $p$, while the lower bound $\underline{r}_{f,\delta}(p)$ is strictly increasing with respect to $p$. After applying simplex saturation, the effective lower bound is non-decreasing and is strictly increasing whenever it is not saturated at $0$.*

**Resolving the Exploration Bottleneck.** Figures 2 and 3 compare BandPO (KL, $\chi^2$, TV) against baselines. Both BandPO and DCPO expand ratio bounds as $p \to 0$, preventing premature suppression of tail actions to facilitate exploration. However, unlike heuristics (DCPO, DAPO) that violate theoretical limits at high probabilities, all BandPO variants maintain strict geometric consistency with the simplex $\Delta^V$. Specifically, BandPO-KL yields the broadest range, while TV and $\chi^2$ correctly capture the simplex saturation at $0$. Thus, BandPO uniquely resolves the exploration bottleneck with a rigorous foundation, guaranteeing valid constraints unlike heuristic approximations.

### 5.5. Solving Band Bounds

For a chosen function $f$ and radius $\delta$, given the probability $p = P(a)$, we solve Problems (16) and (17) to derive the probability ratio bounds $\underline{r}_{f,\delta}(a; P)$ and $\overline{r}_{f,\delta}(a; P)$.

**Simplex Saturation.** For a sufficiently large radius $\delta$, the divergence constraint may extend beyond the boundaries of the probability simplex $\Delta^V$. This creates a conflict between the simplex boundaries and the divergence constraint, rendering Problems (16) and (17) ill-defined. We term this phenomenon **Simplex Saturation**. To enforce the simplex constraint, we formalize this saturation condition and align the Band bounds with the simplex limits, with details deferred to Appendix A.5.

**Proposition 3** (Constraint Saturation). *Based on Inequality (6), denote simplex bounds as $r_{\max} \triangleq 1/p$ and $r_{\min} \triangleq 0$. The optimal Band upper bound $\overline{r}_{f,\delta}(p)$ is given by:*

$$\overline{r}_{f,\delta}(p) = \begin{cases} r_{\max}, & \text{if } g_f(p, r_{\max}) \leq \delta, \\ r^{\dagger}, & \text{otherwise,} \end{cases} \quad (19)$$

*where $r^{\dagger}$ denotes the unique root of $g_f(p, r) = \delta$ in $(1, r_{\max})$, which is guaranteed to exist in the non-saturated regime. Symmetrically, $\underline{r}_{f,\delta}(p) = 0$ if $g_f(p, r_{\min}) \leq \delta$; otherwise, it is the unique root of $g_f(p, r) = \delta$ in $(0, 1)$.*

**Generic Numerical Solver.** In the active regime, the optimal bounds correspond strictly to the unique roots of the binding equation $g_f(p, r) = \delta$. These two roots lie on opposite sides of the stationary point $r = 1$, where $g_f(p, r)$ attains its unique global minimum. Consequently, they can be isolated on the two monotone branches $(r_{\min}, 1)$ and $(1, r_{\max})$, respectively, and efficiently computed via standard bracketed root-finding algorithms, such as bisection method. We provide the rigorous convergence analysis and the standard solver formulation in Appendix A.6, along with a concrete instantiation for the KL-divergence.

**Closed-Form Solutions.** Specific $f$-divergences admit closed-form solutions to Problems (16) and (17), offering superior computational efficiency compared to numerical methods. We provide bounds for TV and Pearson $\chi^2$ divergence, with derivations detailed in Appendix A.7.

### 5.6. BandPO: Band-Constrained Policy Optimization

We propose **BandPO**, a policy optimization framework that substitutes the canonical clipping mechanism of GRPO with the theoretically rigorous Band operator. Formally, BandPO maximizes the objective $\mathcal{J}_{\text{BandPO}}(\theta)$:

$$\mathbb{E}_{x \sim \mathcal{D}, \{y_i\}_{i=1}^G \sim \pi_{\text{old}}} \left[ \frac{1}{G} \sum_{i=1}^G \frac{1}{T_i} \sum_{t=1}^{T_i} \mathcal{J}_t^{\text{Band}}(\theta; y_i) \right]. \quad (20)$$

By employing the probability-aware clipping interval derived in Equation (12), the core per-token surrogate objective $\mathcal{J}_t^{\text{Band}}(\theta; y_i)$ is formulated as:

$$\min \left( r_{t,i} A_{t,i}, \text{Band}_{f,\delta} \left( r_{t,i}; y_{t,i}, \pi_{\text{old}}(\cdot|s_{t,i}) \right) A_{t,i} \right) \\ - \beta D_{\text{KL}}(\pi_\theta \| \pi_{\text{ref}})_t, \quad (21)$$

where $r_{t,i} = \frac{\pi_\theta(y_{t,i}|s_{t,i})}{\pi_{\text{old}}(y_{t,i}|s_{t,i})}$ and the advantage $A_{t,i}$ is computed as described in Section 3.2. By projecting the trust region induced by $f$-divergence into a probability-aware scalar interval specific to the target action $y_{t,i}$, the Band operator strictly confines the ratio within a theoretically feasible domain, thereby balancing optimization stability with the effective exploration of tail strategies.

## 6. Empirical Study

### 6.1. Experimental Setup

**Models and Datasets.** We employ a composite training set merging DAPO (Yu et al., 2025) with MATH Levels 3–5 (Hendrycks et al., 2021) to fine-tune four models spanning diverse architectures and scales: Qwen2.5-3B-Instruct (Qwen et al., 2025) and the DeepSeek-R1-Distill family (Qwen-1.5B/7B, Llama-8B) (DeepSeek-AI et al., 2025). To evaluate reasoning robustness across varying

difficulty levels, we validate on AMC 2023 (MAA, 2023), AIME 2024 (MAA, 2024), and AIME 2025 (MAA, 2025).

**Baselines and Metrics.** To isolate the impact of clipping mechanisms, we implement comparisons within the GRPO framework (Shao et al., 2024). We establish three baselines: (1) **GRPO**, representing canonical symmetric clipping (Schulman et al., 2017); (2) **GRPO w/ Clip-Higher**, representing the state-of-the-art asymmetric clipping introduced by DAPO (Yu et al., 2025); and (3) **GRPO w/ DCPO-DAC**, which replaces fixed clipping with the Dynamic-Adaptive Clipping module from DCPO (Yang et al., 2025). For our method, **GRPO w/ Band$_{\text{KL},\delta}$**, we replace the canonical clipping with the Band. Evaluation metrics include **pass@32** to measure peak reasoning capability (probability of at least one correct solution) and **mean@32** to quantify the expected policy robustness across 32 samples. Unless otherwise specified, BandPO refers to our default implementation, GRPO w/ Band$_{\text{KL},0.05}$.

**Implementation Details.** We conduct all experiments on $8\times$ NVIDIA H200 GPUs using the `verl` framework (Sheng et al., 2024). We train models for 800 steps (for 1.5B/3B) or 500 steps (for 7B/8B), utilizing a global batch size of 256 (mini-batch 64, micro-batch 8) and a learning rate of $1 \times 10^{-6}$. During generation, we set both the sampling temperature and top-$p$ to 1.0. We repeat the experiments over five random seeds and report mean $\pm$ standard deviation. Regarding clipping bounds, GRPO uses the symmetric thresholds with $\epsilon = 0.2$, while Clip-Higher adopts asymmetric thresholds with $\epsilon_+ = 0.28$ and $\epsilon_- = 0.2$. For DCPO-DAC, we use its recommended setting $\epsilon_- = 0.16$ and $\epsilon_+ = 0.2$. For BandPO, we enforce the KL divergence as the trust region constraint with $\delta = 0.05$. Crucially, we implement a CUDA-accelerated parallel bisection method to efficiently solve for the Band bounds.

### 6.2. Main Results

As presented in Table 1, BandPO consistently outperforms all baselines in **mean@32** across diverse models and datasets, demonstrating that the probability-aware trust-region projection improves the expected quality of sampled solutions. Compared with canonical clipping (GRPO), BandPO improves average mean@32 by at least $+1.5$ points across all model scales, with the largest average gain on 3B $(+5.34$ points) and a $+10.46$-point gain on 3B AMC2023. BandPO also outperforms stronger clipping baselines. Relative to Clip-Higher, it improves average mean@32 by $+0.45$, $+2.11$, $+2.41$, and $+0.80$ points across the four model scales. Relative to DCPO-DAC, it improves average mean@32 by $+1.39$, $+0.80$, $+1.14$, and $+0.57$ points. For **pass@32**, BandPO achieves the best average score on the 1.5B, 3B, and 8B models, improving over GRPO by $+5.13$, $+10.38$, and $+3.48$ points, respectively. On the strongest

*Table 1.* **Reasoning performance comparison across model scales (1.5B–8B).** We report mean@32/pass@32 (%) on AMC2023, AIME2024, and AIME2025, with mean $\pm$ std computed over five random seeds. Red highlights the best value in the Average column for either mean@32 or pass@32. GRPO w/ $\mathsf{Band}_{\mathrm{KL},0.05}$ consistently achieves the best average mean@32 across all model scales.

| Method | AMC2023 mean@32/pass@32 | AIME2024 mean@32/pass@32 | AIME2025 mean@32/pass@32 | Average mean@32/pass@32 |
|---|---|---|---|---|
| **DeepSeek-R1-Distill-Qwen-1.5B for 800 training steps** | | | | |
| GRPO | 71.91 ± 0.47 / 93.26 ± 1.20 | 16.71 ± 0.89 / 35.70 ± 3.38 | 22.13 ± 0.52 / 39.40 ± 1.79 | 36.91 ± 0.43 / 56.12 ± 1.00 |
| GRPO w/ Clip-Higher | 76.72 ± 0.29 / 94.38 ± 0.76 | 18.75 ± 0.82 / 42.04 ± 3.39 | 23.69 ± 0.85 / 43.19 ± 2.88 | 39.72 ± 0.51 / 59.87 ± 1.27 |
| GRPO w/ DCPO-DAC | 75.45 ± 0.26 / 93.72 ± 1.25 | 18.13 ± 0.28 / 37.64 ± 2.65 | 22.75 ± 0.92 / 43.16 ± 2.58 | 38.78 ± 0.28 / 58.17 ± 1.26 |
| GRPO w/ Relaxed $\mathsf{Band}_{\mathrm{KL},0.05}$ | 75.05 ± 1.04 / 94.17 ± 0.35 | 18.50 ± 1.04 / 41.33 ± 3.35 | 23.75 ± 0.55 / 44.78 ± 4.33 | 39.10 ± 0.48 / 60.09 ± 1.35 |
| GRPO w/ $\mathsf{Band}_{\mathrm{KL},0.05}$ | 77.56 ± 0.49 / 95.08 ± 0.77 | 18.83 ± 0.81 / 43.54 ± 5.28 | 24.13 ± 0.63 / 45.13 ± 2.94 | 40.17 ± 0.34 / 61.25 ± 0.94 |
| **Qwen2.5-3B-Instruct for 800 training steps** | | | | |
| GRPO | 45.03 ± 1.58 / 73.07 ± 3.56 | 2.81 ± 1.07 / 11.61 ± 2.58 | 2.61 ± 0.79 / 11.14 ± 2.65 | 16.82 ± 1.12 / 31.94 ± 2.10 |
| GRPO w/ Clip-Higher | 51.89 ± 0.79 / 82.15 ± 1.40 | 4.10 ± 0.37 / 14.16 ± 1.17 | 4.15 ± 0.23 / 20.60 ± 3.06 | 20.05 ± 0.33 / 38.97 ± 1.12 |
| GRPO w/ DCPO-DAC | 54.53 ± 0.18 / 85.05 ± 1.46 | 4.69 ± 0.26 / 13.68 ± 2.76 | 4.87 ± 0.15 / 22.83 ± 1.09 | 21.36 ± 0.15 / 40.52 ± 1.32 |
| GRPO w/ Relaxed $\mathsf{Band}_{\mathrm{KL},0.05}$ | 52.52 ± 0.46 / 84.48 ± 1.69 | 4.37 ± 0.82 / 17.02 ± 2.87 | 3.87 ± 0.59 / 18.81 ± 3.09 | 20.25 ± 0.46 / 40.10 ± 1.75 |
| GRPO w/ $\mathsf{Band}_{\mathrm{KL},0.03}$ | 52.45 ± 0.65 / 82.97 ± 3.13 | 3.92 ± 0.33 / 12.56 ± 1.81 | 4.46 ± 0.68 / 23.43 ± 2.98 | 20.28 ± 0.15 / 39.65 ± 1.24 |
| GRPO w/ $\mathsf{Band}_{\mathrm{KL},0.10}$ | 51.45 ± 0.45 / 84.55 ± 1.71 | 3.54 ± 0.39 / 14.35 ± 2.79 | 6.04 ± 0.58 / 21.06 ± 1.95 | 20.35 ± 0.34 / 39.99 ± 1.25 |
| GRPO w/ $\mathsf{Band}_{\mathrm{KL},0.05}$ | 55.49 ± 0.43 / 86.65 ± 1.58 | 4.79 ± 0.17 / 15.36 ± 1.23 | 6.21 ± 0.28 / 24.94 ± 1.73 | 22.16 ± 0.19 / 42.32 ± 0.68 |
| **DeepSeek-R1-Distill-Qwen-7B for 500 training steps** | | | | |
| GRPO | 87.08 ± 0.50 / 95.94 ± 0.86 | 27.98 ± 0.73 / 49.37 ± 1.93 | 31.56 ± 0.80 / 54.95 ± 2.35 | 48.87 ± 0.22 / 66.76 ± 1.48 |
| GRPO w/ Clip-Higher | 87.58 ± 0.61 / 95.00 ± 0.00 | 26.69 ± 0.56 / 46.20 ± 1.75 | 30.65 ± 1.11 / 56.38 ± 3.34 | 48.30 ± 0.36 / 65.86 ± 0.88 |
| GRPO w/ DCPO-DAC | 88.17 ± 0.21 / 95.45 ± 1.00 | 28.83 ± 1.59 / 47.58 ± 2.12 | 31.71 ± 0.94 / 55.90 ± 1.89 | 49.57 ± 0.75 / 66.31 ± 0.57 |
| GRPO w/ Relaxed $\mathsf{Band}_{\mathrm{KL},0.05}$ | 88.45 ± 0.43 / 95.32 ± 0.71 | 27.94 ± 1.16 / 49.54 ± 2.19 | 32.85 ± 0.88 / 54.90 ± 1.78 | 49.75 ± 0.53 / 66.59 ± 0.92 |
| GRPO w/ $\mathsf{Band}_{\mathrm{KL},0.03}$ | 88.15 ± 0.71 / 95.32 ± 0.71 | 28.69 ± 0.73 / 49.87 ± 2.02 | 32.50 ± 0.34 / 55.34 ± 4.30 | 49.78 ± 0.30 / 66.84 ± 1.51 |
| GRPO w/ $\mathsf{Band}_{\mathrm{KL},0.10}$ | 88.94 ± 0.62 / 95.00 ± 0.00 | 29.08 ± 0.96 / 48.82 ± 2.91 | 32.81 ± 0.87 / 57.10 ± 3.53 | 50.28 ± 0.50 / 66.97 ± 1.87 |
| GRPO w/ $\mathsf{Band}_{\mathrm{KL},0.05}$ | 89.50 ± 0.37 / 95.00 ± 0.00 | 28.96 ± 1.12 / 46.28 ± 2.22 | 33.69 ± 1.18 / 56.87 ± 1.70 | 50.71 ± 0.63 / 66.05 ± 0.73 |
| **DeepSeek-R1-Distill-Llama-8B for 500 training steps** | | | | |
| GRPO | 84.98 ± 0.69 / 94.44 ± 0.32 | 23.50 ± 1.14 / 44.99 ± 1.32 | 25.69 ± 1.47 / 51.32 ± 1.96 | 44.72 ± 0.85 / 63.58 ± 0.86 |
| GRPO w/ Clip-Higher | 86.70 ± 0.48 / 94.36 ± 0.44 | 24.79 ± 1.22 / 46.01 ± 0.88 | 27.50 ± 1.33 / 56.72 ± 4.31 | 46.33 ± 0.86 / 65.69 ± 1.57 |
| GRPO w/ DCPO-DAC | 86.44 ± 0.32 / 94.92 ± 0.17 | 23.98 ± 0.65 / 47.52 ± 2.41 | 29.25 ± 0.90 / 53.79 ± 4.67 | 46.56 ± 0.55 / 65.41 ± 1.74 |
| GRPO w/ Relaxed $\mathsf{Band}_{\mathrm{KL},0.05}$ | 86.22 ± 0.69 / 94.85 ± 0.14 | 24.40 ± 0.45 / 49.80 ± 1.51 | 28.93 ± 0.92 / 53.54 ± 4.24 | 46.52 ± 0.27 / 66.06 ± 1.52 |
| GRPO w/ $\mathsf{Band}_{\mathrm{KL},0.05}$ | 87.30 ± 0.64 / 95.22 ± 0.75 | 25.00 ± 0.81 / 49.88 ± 1.17 | 29.08 ± 0.40 / 56.09 ± 3.48 | 47.13 ± 0.29 / 67.06 ± 1.35 |

7B backbone, average pass@32 is similar across methods, while the weaker 3B backbone benefits substantially on AMC2023, with pass@32 increasing from 73.07 to 86.65. Overall, the main comparison supports the central claim that geometry-aware clipping empirically yields a more reliable exploration–exploitation trade-off than fixed and dynamic heuristic clipping bounds.

### 6.3. Heuristic Relaxation Weakens **Band** Performance

Figure 1 shows that $\mathsf{Band}$ enlarges the feasible interval for low-probability actions while imposing tighter constraints in the high-probability regime than fixed asymmetric clipping. We therefore test whether directly relaxing this high-probability side helps, using **GRPO w/ Relaxed Band**$_{\mathrm{KL},0.05}$, which heuristically expands the bound to cover the Clip-Higher range. As shown in Table 1, this relaxation consistently lowers average mean@32 across all model scales. Compared with Relaxed $\mathsf{Band}$, the original $\mathsf{Band}_{\mathrm{KL},0.05}$ improves average mean@32 by +1.07, +1.91, +0.96, and +0.61 points on the 1.5B, 3B, 7B, and 8B models, respectively. The pass@32 trend is similar on 1.5B, 3B, and 8B, while the 7B model shows a small exception under a saturated pass@32 regime. These results indicate that the tighter high-probability constraint is not an undesirable artifact, but a useful consequence of the divergence-induced trust-region projection. Thus, clipping intervals should be derived from the underlying geometry rather than heuristically expanded to match existing clipping ranges.

### 6.4. Trust-Region Radius Sensitivity is Scale-Dependent

BandPO replaces clipping thresholds with a single geometric hyperparameter $\delta$. We study its sensitivity by sweeping $\delta \in \{0.03, 0.05, 0.10\}$ for $\mathsf{Band}_{\mathrm{KL},\delta}$ on the 3B and 7B backbones. As shown in Table 1, the 3B model is more sensitive to this choice: $\delta = 0.05$ achieves the best average mean@32/pass@32 (22.16/42.32), outperforming both $\delta = 0.03$ and $\delta = 0.10$ by about 1.8–1.9 points in mean@32 and 2.3–2.7 points in pass@32. In contrast, the 7B model is much more stable across radii, with average mean@32 varying by less than one point and pass@32 staying within a similarly narrow range. Although $\delta = 0.10$ slightly improves pass@32 on 7B, $\delta = 0.05$ gives the best mean@32, which better reflects expected sample quality. Overall, these results indicate that smaller backbones require more careful trust-region control, while larger backbones are more tolerant to $\delta$. We therefore use $\delta = 0.05$ as a default for $\mathsf{Band}_{\mathrm{KL},\delta}$, rather than a universally optimal constant.

### 6.5. BandPO Unlocks Exploration for Tail Actions

We analyze the clipping dynamics on Qwen2.5-3B to understand the mechanism behind BandPO's performance. While BandPO maintains a high global clipping rate comparable to GRPO (Fig. 4(a)), this metric masks a critical redistribution of the "trust budget": BandPO tightens constraints on high-probability tokens while relaxing them for the tail. **The Bottleneck Resolved:** Fig. 4(b) isolates the critical regime—low-probability actions ($p < 0.2$) with positive

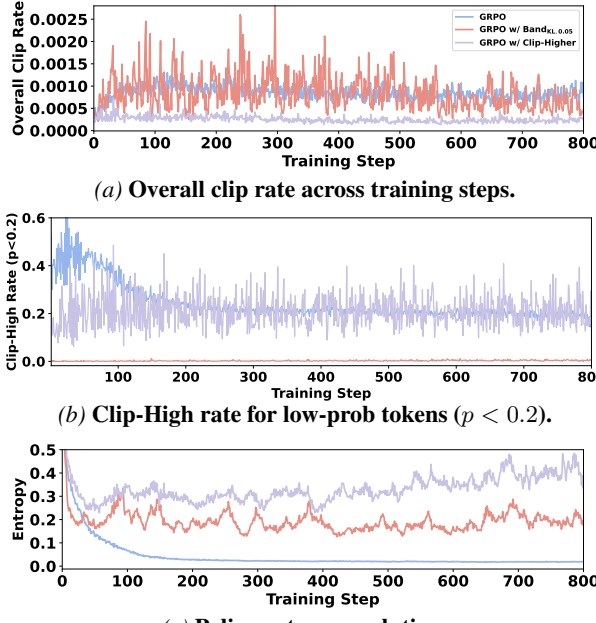

*(a)* **Overall clip rate across training steps.**

*(b)* **Clip-High rate for low-prob tokens** ($p < 0.2$).

*(c)* **Policy entropy evolution.**

*Figure 4.* **Comparison of training dynamics.** (a) Overall clip rate measuring the fraction of clipped tokens relative to total tokens per update. (b) Proportion of clip-high for low-probability tokens ($p < 0.2$) relative to total clipped tokens, identifying erroneous tail-action suppression. (c) Evolution of policy entropy measuring the concentration of action distributions.

advantages. Both GRPO and Clip-Higher erroneously clip 20–60% of these updates, confirming that fixed constants fail to accommodate the geometric scale of tail actions. Conversely, BandPO virtually eliminates this suppression ($\approx 0\%$). **Impact on Entropy:** This structural difference dictates exploration stability. The aggressive tail suppression in GRPO aligns perfectly with its rapid entropy collapse (Fig. 4(c)). By adhering to simplex geometry, BandPO preserves entropy not by merely reducing total clipping, but by unlocking the theoretical margin available to tail strategies, thereby fundamentally resolving the bottleneck.

### 6.6. Wall-Clock Efficiency

We conduct a wall-clock efficiency study on Qwen2.5-3B-Instruct under the same 800-step GRPO training framework used in our main experiments. As shown in Table 2, Band remains efficient end-to-end: it finishes training in 37.56 hours, faster than Clip (39.67 hours) and Higher (39.09 hours). Although its bound computation is more expensive than fixed clipping or DAC, the isolated clipping cost is only 676.3 ms per step, accounting for about 0.4% of the full iteration time. Compared with DAC, Band takes longer in total training time (37.56 vs. 33.81 hours), but this gap is mainly driven by response length rather than clipping overhead. DAC produces substantially shorter responses (1087 tokens on average), whereas Band maintains response lengths close to Clip (1162 vs. 1154 tokens), preserving

*Table 2.* **Training efficiency.** Higher denotes Clip-Higher, and Band denotes $\mathsf{Band}_{\mathrm{KL}, \delta=0.05}$ with the CUDA-parallelized bisection solver. "RespLen" is the average response length. "Total" is the end-to-end training time, "Step" is the full iteration time, and "Clip" isolates the time for clipping bounds in one step.

| Method | RespLen (tok) | Total (hr) | Step (s) | Clip (ms) |
|---|---|---|---|---|
| Clip | 1154±142 | 39.67 | 178.49±17.32 | 5.3±0.2 |
| Higher | 1265±328 | 39.09 | 175.90±37.84 | 5.5±0.1 |
| DAC | 1087±119 | 33.81 | 152.15±13.63 | 51.0±3.1 |
| Band | 1162±152 | 37.56 | 169.04±17.36 | 676.3±6.6 |

longer reasoning trajectories for difficult math problems. Thus, Band introduces negligible step-level overhead while retaining the practical efficiency of clipping-based RL.

## 7. Discussion

**Limitations.** Our empirical evaluation focuses on mathematical reasoning tasks with sparse, verifiable rewards. While this setting directly matches the exploration bottleneck studied in this work, it does not fully cover broader LLM RL scenarios, such as code generation or tasks with denser process-level rewards. In addition, due to hardware limitations, we did not evaluate the Band operator on ultra-large models (72B+ parameters), though its computational complexity scales with sequence length rather than model depth, suggesting generalizability to larger architectures. For given $\delta$ and $f$-divergence, Band bounds can be precomputed for $O(1)$ retrieval; our current implementation employs bisection-based numerical solvers for robustness without pre-computation overhead.

**Future Work.** Future research will explore alternative metrics for bound computation, such as token-level entropy, to capture distributional shifts from different perspectives and reveal more effective mechanisms for policy optimization.

## 8. Conclusion

We introduced BandPO, a principled optimization framework that bridges the gap between computationally efficient clipping and rigorous trust region constraints. By projecting $f$-divergence balls into dynamic, probability-aware bounds, BandPO effectively resolves the exploration bottleneck for high-advantage tail actions while maintaining training stability. Extensive experiments across models from 1.5B to 8B demonstrate that BandPO significantly outperforms heuristic baselines and robustly prevents entropy collapse. These results highlight the limitations of static heuristics and demonstrate the efficacy of geometrically grounded constraints in optimizing complex reasoning policies.

## Acknowledgements

We sincerely thank the anonymous reviewers, meta-reviewer, Area Chairs, and Program Chairs for their thoughtful feedback and constructive suggestions. Their comments helped us strengthen the empirical validation and better clarify the practical scope of the method. This work was supported by the National Natural Science Foundation of China (No. 62236004).

## Impact Statement

This paper presents work whose goal is to advance the field of Machine Learning. There are many potential societal consequences of our work, none of which we feel must be specifically highlighted here.

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

# A. Omitted Proofs

## A.1. Proof of Lemma 1

*Proof.* Without loss of generality, we analyze the structural properties of the optimal solution for the maximization formulation in Problem (10). The derivation for the minimization case is identical due to the shared feasible set.

Let $q \triangleq Q(a)$ be a fixed probability mass assigned to the target action $a$. To maximize the feasible range of $q$ under the divergence constraint, the optimal policy must distribute the remaining probability mass $1 - q$ over the complement set $\mathcal{V}' = \mathcal{V} \setminus \{a\}$ such that the divergence contribution is minimized. This sub-problem is formulated as:

$$\min_{\{Q(b)\}_{b \in \mathcal{V}'}} \quad \sum_{b \in \mathcal{V}'} P(b) f \left( \frac{Q(b)}{P(b)} \right) \quad \text{s.t.} \quad \sum_{b \in \mathcal{V}'} Q(b) = 1 - q. \tag{22}$$

**1. Optimality via Symmetry and Convexity.** Observe that the objective function is a sum of strictly convex terms, and the variables $\{Q(b)\}_{b \in \mathcal{V}'}$ appear symmetrically in the objective (weighted by $P(b)$) and the constraint. Due to the strict convexity of the generator function $f$, the global minimum of this sub-problem must satisfy the symmetry condition where the likelihood ratio is constant across all complement actions. Formally, for any $b_i, b_j \in \mathcal{V}'$, optimality requires:

$$\frac{Q(b_i)}{P(b_i)} = \frac{Q(b_j)}{P(b_j)} = c, \tag{23}$$

for some scalar $c \geq 0$. If this were not true, one could construct a strictly better solution by averaging the ratios, thereby reducing the strictly convex objective (Jensen's Inequality). Thus, the optimal structure is necessarily $Q(b) = cP(b)$ for all $b \neq a$.

**2. Determination of the Scaling Factor.** Substituting $Q(b) = cP(b)$ into the probability mass constraint $\sum_{b \in \mathcal{V}'} Q(b) = 1 - q$ yields:

$$c \sum_{b \in \mathcal{V}'} P(b) = 1 - q \implies c(1 - P(a)) = 1 - q \implies c = \frac{1 - q}{1 - P(a)}. \tag{24}$$

Recall that $q = rP(a)$ where $r$ is the ratio at the target action. This recovers the scaling factor $c(r) = \frac{1 - rP(a)}{1 - P(a)}$.

**3. Strict Interiority.** In the context of LLM post-training, the reference policy $\pi_{\text{old}}$ is derived from a Softmax distribution, ensuring $P(v) > 0$ for all $v \in \mathcal{V}$. Furthermore, the trust-region radius $\delta$ is typically small (e.g., $\delta \ll D_f(\mathbb{I}_a \| P)$), ensuring that the feasible set does not include the degenerate solution where $Q(a) = 1$ (which would imply an infinite or maximally large divergence). Consequently, we strictly have $q < 1$, which implies $c = \frac{1 - q}{1 - P(a)} > 0$. Since $P(b) > 0$ and $c > 0$, it follows that $Q(b) > 0$ for all $b$. Thus, the optimal solution strictly resides in the interior of the probability simplex, justifying the validity of the derivation without requiring ad-hoc assumptions. $\square$

## A.2. Proof of Theorem 1

*Proof.* We aim to prove that the scalar function $g_f(p, r)$ is strictly convex and that the optimal clipping bounds are determined by either the unique active roots of $g_f(p, r) = \delta$ or, when the divergence constraint is inactive on a branch, by the corresponding simplex boundary.

**1. Strict Convexity of $g_f(p, r)$.** Recall the definition of the scalarized divergence function:

$$g_f(p, r) = pf(r) + (1 - p)f(c(r)), \quad \text{where } c(r) = \frac{1 - rp}{1 - p}. \tag{25}$$

We compute the first and second derivatives with respect to $r$ on the active interior $r \in (0, 1/p)$. First, note that $\frac{dc(r)}{dr} = \frac{-p}{1 - p}$. The first derivative is:

$$\frac{\partial g_f}{\partial r} = pf'(r) + (1 - p)f'(c(r)) \left( \frac{-p}{1 - p} \right) = p \left[ f'(r) - f'(c(r)) \right]. \tag{26}$$

On the active interior $r \in (0, 1/p)$, both $r$ and $c(r)$ lie in $\mathbb{R}_{++}$. Where the derivatives exist, the second derivative is

$$\frac{\partial^2 g_f}{\partial r^2} = pf''(r) + \frac{p^2}{1-p}f''(c(r)). \tag{27}$$

Since $f$ is convex, this quantity is nonnegative. More importantly, strict convexity of $g_f$ follows directly from composition: the map $r \mapsto pf(r)$ is strictly convex on the active interior because $p > 0$ and $f$ is strictly convex, while the complement term $r \mapsto (1-p)f(c(r))$ is convex because $c(r)$ is affine and $f$ is convex. The sum of a strictly convex function and a convex function is strictly convex. Therefore, $g_f(p, r)$ is strictly convex on $(0, 1/p)$, and by lower-semicontinuous extension the endpoint characterization applies on $[0, 1/p]$.

**2. Global Minimum.** We examine the critical point where $\frac{\partial g_f}{\partial r} = 0$. $p[f'(r) - f'(c(r))] = 0 \implies f'(r) = f'(c(r))$. Since $f$ is strictly convex, $f'$ is strictly increasing and injective, implying $r = c(r)$. Solving $r = \frac{1-rp}{1-p}$ yields $r(1-p) = 1 - rp \implies r = 1$. At $r = 1$, we have $g_f(p, 1) = pf(1) + (1-p)f(1) = 0$ (since $f(1) = 0$). Therefore, $g_f$ achieves its unique global minimum of $0$ at $r = 1$.

**3. Existence and Uniqueness of Active Roots under Simplex Saturation.** Since $g_f(p, r)$ is strictly convex and attains its unique minimum at $r = 1$, it is strictly decreasing on the interval $[0, 1)$ and strictly increasing on $(1, 1/p]$, with boundary values interpreted as one-sided extended limits when $f(0) = +\infty$. For any trust-region radius $\delta > 0$, we have $g_f(p, 1) = 0 < \delta$. The scalar feasible domain induced by the simplex constraint is $r \in [0, 1/p]$.

We distinguish between active roots and simplex-saturated solutions:

- **Lower branch.** On $[0, 1]$, if the boundary value $g_f(p, 0)$, interpreted as a one-sided extended limit, satisfies $g_f(p, 0) > \delta$, then continuity on $(0, 1)$ and strict monotonicity guarantee exactly one root $\underline{r}_{f,\delta}(p) \in (0, 1)$ satisfying

$$g_f(p, \underline{r}_{f,\delta}(p)) = \delta. \tag{28}$$

  The uniqueness follows from the strict monotonicity of $g_f(p, r)$ on $[0, 1)$. If instead $g_f(p, 0) \leq \delta$, then the entire lower branch is feasible and the lower bound saturates at the simplex boundary:

$$\underline{r}_{f,\delta}(p) = 0. \tag{29}$$

- **Upper branch.** On $[1, 1/p]$, if the boundary value $g_f(p, 1/p)$, interpreted as a one-sided extended limit, satisfies $g_f(p, 1/p) > \delta$, then continuity on $(1, 1/p)$ and strict monotonicity guarantee exactly one root $\overline{r}_{f,\delta}(p) \in (1, 1/p)$ satisfying:

$$g_f(p, \overline{r}_{f,\delta}(p)) = \delta. \tag{30}$$

  The uniqueness follows from the strict monotonicity of $g_f(p, r)$ on $(1, 1/p]$. If instead $g_f(p, 1/p) \leq \delta$, then the entire upper branch is feasible and the upper bound saturates at the simplex boundary:

$$\overline{r}_{f,\delta}(p) = 1/p. \tag{31}$$

Thus, the inequality constraint $D_f(Q\|P) \leq \delta$, which is equivalent to $g_f(p, r) \leq \delta$ over the scalar simplex domain $r \in [0, 1/p]$, corresponds to a feasible interval containing $r = 1$. Since the objective is to maximize or minimize $r$, the solution must lie at an endpoint of this feasible interval. In the active regime, the endpoint is the unique root of the binding equation $g_f(p, r) = \delta$; in the saturated regime, the endpoint coincides with the corresponding simplex boundary. $\qquad\square$

### A.3. Proof of Proposition 1 (Asymptotic Behavior)

*Proof.* We investigate the limiting behavior of the effective bounds $\overline{r}_{f,\delta}(p)$ and $\underline{r}_{f,\delta}(p)$ under the scalar feasible set $g_f(p, r) \leq \delta$ and the simplex constraint $r \in [0, 1/p]$. Let

$$c_p(r) \triangleq \frac{1-rp}{1-p}, \qquad g_f(p, r) = pf(r) + (1-p)f(c_p(r)). \tag{32}$$

**1. Tail Expansion ($p \to 0^+$).** We first show that the upper bound diverges. Fix any finite $R > 1$. For sufficiently small $p$, we have $R < 1/p$, so $R$ lies inside the simplex domain. Moreover, $c_p(R) \to 1$ as $p \to 0^+$. Since $f(1) = 0$ and $R$ is fixed, we obtain

$$g_f(p, R) = pf(R) + (1-p)f(c_p(R)) \to 0. \tag{33}$$

Therefore, for sufficiently small $p$, $R$ is feasible under $g_f(p, R) \leq \delta$. This implies $\overline{r}_{f,\delta}(p) \geq R$ for sufficiently small $p$. Because $R > 1$ is arbitrary, we conclude that

$$\lim_{p \to 0^+} \overline{r}_{f,\delta}(p) = +\infty. \tag{34}$$

We next show that the lower bound converges to 0. Fix any $\eta \in (0, 1)$. Again, $c_p(\eta) \to 1$ as $p \to 0^+$, and hence

$$g_f(p, \eta) = pf(\eta) + (1 - p)f(c_p(\eta)) \to 0. \tag{35}$$

Thus, for sufficiently small $p$, $\eta$ is feasible. By the definition of the lower endpoint,

$$0 \leq \underline{r}_{f,\delta}(p) \leq \eta \tag{36}$$

for all sufficiently small $p$. Since $\eta \in (0, 1)$ is arbitrary, we obtain

$$\lim_{p \to 0^+} \underline{r}_{f,\delta}(p) = 0. \tag{37}$$

This argument does not require $f(0)$ to be finite, and therefore also covers the KL generator used in this paper.

**2. Head Constriction of the Upper Bound ($p \to 1^-$).** For every feasible distribution $Q$, we have $Q(a) = rP(a) = rp \leq 1$. Therefore every feasible ratio satisfies

$$r \leq \frac{1}{p}. \tag{38}$$

Since the upper endpoint lies on the branch $r \geq 1$, we have

$$1 \leq \overline{r}_{f,\delta}(p) \leq \frac{1}{p}. \tag{39}$$

Taking $p \to 1^-$ yields

$$\lim_{p \to 1^-} \overline{r}_{f,\delta}(p) = 1. \tag{40}$$

**3. Divergence-dependent Behavior of the Lower Bound as $p \to 1^-$.** The lower bound is not squeezed by the simplex upper boundary in the same way. It lies on the branch $r \in [0, 1]$, and the interval $[0, 1]$ does not collapse as $p \to 1^-$. Hence, the limit of $\underline{r}_{f,\delta}(p)$ is not universal.

Assume the extended asymptotic slope

$$C_\infty \triangleq \lim_{u \to +\infty} \frac{f(u)}{u} \tag{41}$$

exists, allowing $C_\infty = +\infty$. For any fixed $r \in [0, 1)$, as $p \to 1^-$, we have $c_p(r) \to +\infty$. Moreover,

$$
\begin{aligned}
(1 - p)f(c_p(r)) &= (1 - p)c_p(r)\frac{f(c_p(r))}{c_p(r)} \\
&= (1 - p + p(1 - r))\frac{f(c_p(r))}{c_p(r)} \to (1 - r)C_\infty.
\end{aligned}
\tag{42}
$$

Therefore, the limiting lower-branch constraint is governed by

$$h_f(r) \triangleq f(r) + (1 - r)C_\infty, \qquad r \in [0, 1), \tag{43}$$

with $h_f(1) \triangleq 0$. Since $h_f(r)$ is non-increasing on $[0, 1]$ for convex $f$ with asymptotic slope $C_\infty$, the limiting feasible lower branch is an interval of the form $[r^*, 1]$. Therefore, the head-limit lower endpoint is characterized by

$$r^* = \inf \left\{ r \in [0, 1] \ : \ h_f(r) \leq \delta \right\}. \tag{44}$$

When the limiting lower branch is active and non-saturated, this characterization reduces to the binding equation

$$f(r^*) + (1 - r^*)C_\infty = \delta. \tag{45}$$

If $h_f(0) \le \delta$, the lower branch is saturated and $r^* = 0$.

For the divergences considered in this paper, where KL refers to the old-policy-anchored forward KL induced by $f_{\mathrm{KL}}(u) = -\log u + u - 1$, this yields

$$r^*_{\mathrm{KL}} = e^{-\delta}, \qquad r^*_{\mathrm{TV}} = \max\{1 - \delta, 0\}, \qquad r^*_{\chi^2} = 1. \tag{46}$$

Thus, the three limits stated in Proposition 1 are universal, whereas the head-regime limit of the lower bound is divergence-dependent and therefore is not included as a universal statement. This completes the proof. $\qquad\square$

### A.4. Proof of Proposition 2 (Monotonicity)

*Proof.* We establish the monotonicity of the effective clipping bounds by separating the active non-saturated roots from simplex-saturated endpoints. First, consider an active interior root of

$$F(p, r) \triangleq g_f(p, r) - \delta = 0, \tag{47}$$

where $r \in (0, 1/p)$ and the corresponding endpoint is not saturated at the simplex boundary. For such an active root, the Implicit Function Theorem gives

$$\frac{dr}{dp} = -\frac{\partial F/\partial p}{\partial F/\partial r}. \tag{48}$$

**1. Positivity of $\frac{\partial F}{\partial p}$ via Bregman Divergence.** Recall that

$$c \triangleq c_p(r) = \frac{1 - rp}{1 - p}. \tag{49}$$

Differentiating $g_f(p, r)$ with respect to $p$, and noting that

$$\frac{\partial c}{\partial p} = \frac{1 - r}{(1 - p)^2}, \tag{50}$$

we obtain

$$\begin{aligned}
\frac{\partial F}{\partial p} &= f(r) - f(c) + (1 - p)f'(c)\frac{\partial c}{\partial p} \\
&= f(r) - f(c) + f'(c)\frac{1 - r}{1 - p}.
\end{aligned} \tag{51}$$

Since

$$c - r = \frac{1 - rp}{1 - p} - r = \frac{1 - r}{1 - p}, \tag{52}$$

Equation (51) becomes

$$\frac{\partial F}{\partial p} = f(r) - f(c) - f'(c)(r - c). \tag{53}$$

The right-hand side is the scalar Bregman divergence generated by $f$, evaluated between $r$ and $c$. For strictly convex differentiable $f$, this quantity is strictly positive whenever $r \ne c$. On an active clipping endpoint, $r \ne 1$ because $\delta > 0$, and $r = c$ would imply $r = 1$. Therefore,

$$\frac{\partial F}{\partial p} > 0 \tag{54}$$

for every active non-saturated root.

**2. Sign of $\frac{\partial F}{\partial r}$.** Differentiating $F$ with respect to $r$ yields

$$\frac{\partial F}{\partial r} = pf'(r) + (1 - p)f'(c)\left(\frac{-p}{1 - p}\right) = p\left[f'(r) - f'(c)\right]. \tag{55}$$

Since $f$ is strictly convex, $f'$ is strictly increasing. We analyze the two active branches separately.

- **Upper active root.** For the upper root, $r > 1$. Then

$$c = \frac{1 - rp}{1 - p} < 1, \tag{56}$$

and therefore $r > c$. Hence $f'(r) > f'(c)$ and

$$\left.\frac{\partial F}{\partial r}\right|_{\bar{r}} > 0. \tag{57}$$

- **Lower active root.** For the lower root, $r < 1$. Then

$$c = \frac{1 - rp}{1 - p} > 1, \tag{58}$$

and therefore $r < c$. Hence $f'(r) < f'(c)$ and

$$\left.\frac{\partial F}{\partial r}\right|_{\underline{r}} < 0. \tag{59}$$

**3. Active-root Monotonicity.** Combining Equations (48), (54), (57), and (59), we obtain

$$\frac{d\bar{r}_{f,\delta}(p)}{dp} = -\frac{(+)}{(+)} < 0, \tag{60}$$

$$\frac{d\underline{r}_{f,\delta}(p)}{dp} = -\frac{(+)}{(-)} > 0. \tag{61}$$

Thus, in the active non-saturated regime, the upper bound is strictly decreasing in $p$, while the lower bound is strictly increasing in $p$.

**4. Effect of Simplex Saturation.** It remains to handle the cases where the active root does not exist because the entire branch is feasible under the trust-region constraint.

For the upper branch, saturation gives

$$\bar{r}_{f,\delta}(p) = \frac{1}{p}, \tag{62}$$

which is strictly decreasing in $p$ on any saturated interval. Together with the active-root result in Equation (60), this shows that the upper clipping bound decreases with $p$ both in the active regime and in the simplex-saturated regime.

For the lower branch, saturation gives

$$\underline{r}_{f,\delta}(p) = 0, \tag{63}$$

which is constant and hence non-decreasing. When the lower endpoint is active, Equation (61) shows that it is strictly increasing. Consequently, after applying simplex saturation, the effective lower clipping bound is globally non-decreasing and is strictly increasing whenever it is not saturated at $0$.

The argument above applies to strictly convex differentiable generators. For the TV generator, which is convex but not strictly convex and is not differentiable at $1$, the same monotonicity conclusion follows directly from its closed-form bounds in Appendix A.7. This completes the proof. $\square$

### A.5. Proof of Proposition 3 (Constraint Activity and Saturation)

*Proof.* We analyze the existence of interior roots for the binding equation $g_f(p, r) = \delta$ relative to the simplex boundaries. Recall from Theorem 1 that $g_f(p, r)$ is strictly convex with a global minimum $g_f(p, 1) = 0$.

**Upper Bound Saturation.** Consider the function $g_f(p, r)$ on the interval $[1, r_{\max}]$, where $r_{\max} = 1/p$. Since $g_f$ is strictly increasing for $r > 1$, the maximum divergence is attained at the boundary: $g_{\text{peak}} \triangleq g_f(p, r_{\max})$.

- **Inactive Regime ($g_{\text{peak}} \leq \delta$):** If the maximal divergence is within the trust region, then $g_f(p, r) \leq g_{\text{peak}} \leq \delta$ holds for all $r \in [1, r_{\max}]$. The constraint is inactive. Since the objective in Eq. (10) is to maximize the ratio, the optimal solution saturates at the boundary: $\bar{r}_{f,\delta}(p) = r_{\max}$. In this case, no interior root exists for $g_f(p, r) = \delta$ (unless $\delta = g_{\text{peak}}$, where the root is the boundary itself).

- **Active Regime ($g_{\text{peak}} > \delta$):** Here we have $g_f(p, 1) = 0 < \delta$ and $g_f(p, r_{\max}) > \delta$. Since $g_f$ is continuous on $[1, r_{\max}]$, the Intermediate Value Theorem guarantees the existence of a root $r^\dagger \in (1, r_{\max})$ such that $g_f(p, r^\dagger) = \delta$. Furthermore, due to the strict monotonicity of $g_f$ on this interval, this root is unique.

**Lower Bound Saturation.** Symmetrically, consider the interval $[r_{\min}, 1]$ where $r_{\min} = 0$. $g_f(p, r)$ is strictly decreasing on this interval.

- **Inactive Regime ($g_f(p, r_{\min}) \leq \delta$):** The constraint $g_f(p, r) \leq \delta$ holds for all $r \in [0, 1]$. Minimizing $r$ leads to saturation at the boundary: $\underline{r}_{f,\delta}(p) = 0$.

- **Active Regime ($g_f(p, r_{\min}) > \delta$):** Since $g_f(p, 1) = 0 < \delta$ and $g_f(p, 0) > \delta$, there exists a unique root in $(0, 1)$.

Combining these cases yields the formulation in Proposition 3. $\qquad\square$

### A.6. Numerical Implementation Details

This section details the numerical procedures for solving the Band bounds in the general case where closed-form solutions are unavailable (e.g., KL divergence).

#### A.6.1. CONVERGENCE GUARANTEES

We first establish the theoretical basis for the global convergence of bracketed root-finding methods on the scalarized constraint function.

**Lemma 2** (Strict Monotonicity on Intervals). *Let $f : \mathbb{R}_+ \to \mathbb{R} \cup \{+\infty\}$ be strictly convex, with $f(1) = 0$, and differentiable on $\mathbb{R}_{++}$. For any $p \in (0, 1)$, define the scalarized constraint function*

$$g_f(p, r) \triangleq pf(r) + (1 - p)f\left(\frac{1 - rp}{1 - p}\right). \tag{64}$$

*On the active interior $r \in (0, 1/p)$, $g_f(p, r)$ is strictly convex with respect to $r$, attains its unique minimum at $r = 1$, and is strictly decreasing on $(0, 1)$ and strictly increasing on $(1, 1/p)$. Consequently, on each active non-saturated branch, the equation $g_f(p, r) = \delta$ has at most one root.*

*Proof.* Let

$$c(r) \triangleq \frac{1 - rp}{1 - p}. \tag{65}$$

On the active interior $r \in (0, 1/p)$, both $r$ and $c(r)$ lie in $\mathbb{R}_{++}$. The strict convexity of $g_f$ follows from the strict convexity of the target term $r \mapsto pf(r)$ and the convexity of the complement term $r \mapsto (1 - p)f(c(r))$, where $c(r)$ is affine. Moreover,

$$\frac{\partial g_f(p, r)}{\partial r} = p\left[f'(r) - f'(c(r))\right]. \tag{66}$$

Since $f$ is strictly convex, $f'$ is strictly increasing. For $r < 1$, we have $c(r) > 1$, hence $r < c(r)$ and

$$\frac{\partial g_f(p, r)}{\partial r} < 0. \tag{67}$$

For $r > 1$, we have $c(r) < 1$, hence $r > c(r)$ and

$$\frac{\partial g_f(p, r)}{\partial r} > 0. \tag{68}$$

Therefore, $g_f(p, r)$ is strictly decreasing on $(0, 1)$ and strictly increasing on $(1, 1/p)$. This establishes the branch-wise monotonicity required for root isolation and bisection in the active non-saturated regime.

For the TV generator, $f_{\text{TV}}(u) = \frac{1}{2}|u - 1|$ is convex but neither strictly convex nor differentiable at $u = 1$. Nevertheless, its scalarized constraint admits the closed form

$$g_{\text{TV}}(p, r) = p|r - 1|. \tag{69}$$

Hence $g_{\mathrm{TV}}(p, r)$ is strictly decreasing on $(0, 1)$ and strictly increasing on $(1, 1/p)$. Thus, the same root-isolation and bisection logic applies to TV as a special closed-form case. $\qquad\square$

**Theorem 2** (Global Convergence of Bisection). *Let $r_{\min} \triangleq 0$ and $r_{\max} \triangleq 1/p$, with endpoint values interpreted as one-sided extended limits when necessary. Consider the active non-saturated regime where*

$$g_f(p, r_{\min}) > \delta \quad and \quad g_f(p, r_{\max}) > \delta. \tag{70}$$

*Then the equation $g_f(p, r) = \delta$ has exactly two roots: a lower root $\underline{r} \in (r_{\min}, 1)$ and an upper root $\overline{r} \in (1, r_{\max})$. Applying bisection on the brackets $[r_{\min}, 1]$ and $[1, r_{\max}]$ guarantees linear convergence to $\underline{r}$ and $\overline{r}$, respectively, to arbitrary precision $\epsilon$.*

*Proof.* By Lemma 2, $g_f(p, r)$ is continuous and strictly monotonic on each active interior bracket. Moreover, $g_f(p, 1) = 0 < \delta$. Under the active condition in Equation (70), the boundary values on both branches exceed $\delta$. Thus, the Intermediate Value Theorem guarantees the existence of one root on each branch, and strict monotonicity guarantees uniqueness. The standard convergence analysis of the bisection method then gives linear convergence on both brackets. $\qquad\square$

### A.6.2. STANDARD BISECTION SOLVER ALGORITHMS

We present the standard bisection procedure to solve $g_f(p, r) - \delta = 0$. Let $\epsilon$ be the convergence tolerance (e.g., $10^{-6}$).

**Upper Bound Solver ($\overline{r}$).** First check whether the upper branch is simplex-saturated. If

$$g_f\left(p, \frac{1}{p}\right) \le \delta, \tag{71}$$

with $g_f(p, 1/p)$ interpreted as a one-sided extended limit when necessary, then return $\overline{r}_{f,\delta}(p) = 1/p$. Otherwise, the active upper root lies in $(1, 1/p)$, and the objective function $h(r) \triangleq g_f(p, r) - \delta$ is strictly **increasing** on the active bracket.

- **Initialize:** $L \leftarrow 1$, $R \leftarrow 1/p$.
- **Iterate** until $|R - L| < \epsilon$:
$$m \leftarrow \frac{L + R}{2}; \quad \text{if } g_f(p, m) \le \delta, \text{ set } L \leftarrow m; \text{ else } R \leftarrow m. \tag{72}$$
- **Return:** $L$ (approximating $\overline{r}_{f,\delta}$).

**Lower Bound Solver ($\underline{r}$).** First check whether the lower branch is simplex-saturated. If

$$g_f(p, 0) \le \delta, \tag{73}$$

with $g_f(p, 0)$ interpreted as a one-sided extended limit when necessary, then return $\underline{r}_{f,\delta}(p) = 0$. Otherwise, the active lower root lies in $(0, 1)$, and the objective function $h(r) \triangleq g_f(p, r) - \delta$ is strictly **decreasing** on the active bracket. Note the reversed conditional update logic compared to the upper bound.

- **Initialize:** $L \leftarrow 0$, $R \leftarrow 1$.
- **Iterate** until $|R - L| < \epsilon$:
$$m \leftarrow \frac{L + R}{2}; \quad \text{if } g_f(p, m) \le \delta, \text{ set } R \leftarrow m; \text{ else } L \leftarrow m. \tag{74}$$
- **Return:** $R$ (approximating $\underline{r}_{f,\delta}$).

### A.6.3. INSTANTIATION: FORWARD KL DIVERGENCE

We illustrate the solver with the forward KL divergence used in our implementation. Throughout this paper, the forward KL trust region is interpreted in the old-policy-anchored sense: the reference policy $P$ is the anchor and provides the weighting distribution, while $Q$ denotes the candidate policy. Under the $f$-divergence convention in Equation (7),

$$D_f(Q\|P) = \sum_a P(a) f\left(\frac{Q(a)}{P(a)}\right), \tag{75}$$

choosing

$$f_{\mathrm{KL}}(u) = -\log u + u - 1 \tag{76}$$

yields

$$
\begin{aligned}
D_f(Q\|P) &= \sum_a P(a) \left[ -\log \frac{Q(a)}{P(a)} + \frac{Q(a)}{P(a)} - 1 \right] \\
&= \sum_a P(a) \log \frac{P(a)}{Q(a)} = D_{\mathrm{KL}}(P\|Q).
\end{aligned}
\tag{77}
$$

Thus, although the scalarized constraint is written as $D_f(Q\|P)$, the KL instantiated here is the old-policy-weighted forward KL trust region $D_{\mathrm{KL}}(P\|Q)$.

Substituting $f_{\mathrm{KL}}(u) = -\log u + u - 1$ into the scalarized form yields the binding equation:

$$p(-\log r + r - 1) + (1 - p)(-\log c(r) + c(r) - 1) = \delta, \tag{78}$$

where

$$c(r) = \frac{1 - rp}{1 - p} \tag{79}$$

is the complement scaling factor. The linear terms cancel exactly:

$$p(r - 1) + (1 - p)(c(r) - 1) = 0. \tag{80}$$

Therefore, the binding equation reduces to

$$-p \log r - (1 - p) \log \left( \frac{1 - rp}{1 - p} \right) = \delta. \tag{81}$$

Equivalently, this is the binary forward KL constraint between the old target/complement masses $(p, 1 - p)$ and the candidate masses $(rp, 1 - rp)$:

$$p \log \frac{p}{rp} + (1 - p) \log \frac{1 - p}{1 - rp} = \delta. \tag{82}$$

This equation involves $r$ both in the target logarithmic term and inside the complement mass $1 - rp$, and does not admit a simple closed-form solution in elementary functions. Therefore, the generic bisection solver described in Appendix A.6.2 is used to compute the exact Band bounds for forward-KL-based trust regions.

## A.7. Derivation of Closed-form Band Bounds for TV and Pearson $\chi^2$

*Proof.* Recall from Theorem 1 that the effective bounds are the endpoints of the scalar feasible interval induced by $g_f(p, r) \le \delta$ over the simplex domain $r \in [0, 1/p]$. In the active non-saturated regime, these endpoints are the unique roots of the scalar binding equation; otherwise, they saturate at the corresponding simplex boundaries. Therefore, in this proof we first derive the raw roots of the binding equation

$$g_f(p, r) \triangleq pf(r) + (1 - p)f\left( \frac{1 - rp}{1 - p} \right) = \delta. \tag{83}$$

We then obtain the final effective bounds by intersecting these raw roots with the simplex interval $[0, 1/p]$. We substitute the specific generator functions for Total Variation and Pearson $\chi^2$ to derive the raw roots.

**1. Total Variation (TV).** The generator function is defined as $f_{\mathrm{TV}}(u) = \frac{1}{2}|u - 1|$. Substituting this into Eq. (83) gives:

$$g_{\mathrm{TV}}(p, r) = \frac{p}{2}|r - 1| + \frac{1 - p}{2}\left| \frac{1 - rp}{1 - p} - 1 \right| = \delta. \tag{84}$$

We simplify the complement term. Note that:

$$\frac{1 - rp}{1 - p} - 1 = \frac{1 - rp - (1 - p)}{1 - p} = \frac{p(1 - r)}{1 - p}. \tag{85}$$

Thus, the second term in the binding equation simplifies to:

$$\frac{1-p}{2} \left| \frac{p(1-r)}{1-p} \right| = \frac{1-p}{2} \cdot \frac{p}{1-p} |1-r| = \frac{p}{2}|r-1|. \tag{86}$$

Substituting this back yields a unified expression:

$$g_{\mathrm{TV}}(p,r) = \frac{p}{2}|r-1| + \frac{p}{2}|r-1| = p|r-1|. \tag{87}$$

Setting $g_{\mathrm{TV}}(p,r) = \delta$, we solve for $r$:

$$|r-1| = \frac{\delta}{p} \implies r = 1 \pm \frac{\delta}{p}. \tag{88}$$

**2. Pearson $\chi^2$ Divergence.** The generator function is $f_{\chi^2}(u) = (u-1)^2$. Substituting this into Eq. (83):

$$g_{\chi^2}(p,r) = p(r-1)^2 + (1-p)\left(\frac{1-rp}{1-p} - 1\right)^2 = \delta. \tag{89}$$

Using the same algebraic simplification for the complement term as in the TV case:

$$\left(\frac{1-rp}{1-p} - 1\right)^2 = \left(\frac{p(1-r)}{1-p}\right)^2 = \frac{p^2}{(1-p)^2}(r-1)^2. \tag{90}$$

Substituting this back into $g_{\chi^2}(p,r)$:

$$g_{\chi^2}(p,r) = p(r-1)^2 + (1-p)\frac{p^2}{(1-p)^2}(r-1)^2 \tag{91}$$

$$= p(r-1)^2 \left[1 + \frac{p}{1-p}\right] \tag{92}$$

$$= p(r-1)^2 \left[\frac{1-p+p}{1-p}\right] \tag{93}$$

$$= \frac{p}{1-p}(r-1)^2. \tag{94}$$

Setting $g_{\chi^2}(p,r) = \delta$ and solving for $r$:

$$(r-1)^2 = \delta\frac{1-p}{p} \implies r = 1 \pm \sqrt{\frac{\delta(1-p)}{p}}. \tag{95}$$

**Practical Implementation (Simplex Constraints).** The expressions derived above are the raw roots of the binding equation $g_f(p,r) = \delta$. They must still be intersected with the scalar simplex domain $r \in [0, 1/p]$ to obtain the effective clipping endpoints. To avoid ambiguity, denote these raw roots by $\widetilde{\underline{r}}_{f,\delta}(p)$ and $\widetilde{\overline{r}}_{f,\delta}(p)$. For TV, the raw roots are

$$\widetilde{\underline{r}}_{\mathrm{TV},\delta}(p) = 1 - \frac{\delta}{p}, \qquad \widetilde{\overline{r}}_{\mathrm{TV},\delta}(p) = 1 + \frac{\delta}{p}. \tag{96}$$

For Pearson $\chi^2$, the raw roots are

$$\widetilde{\underline{r}}_{\chi^2,\delta}(p) = 1 - \sqrt{\frac{\delta(1-p)}{p}}, \qquad \widetilde{\overline{r}}_{\chi^2,\delta}(p) = 1 + \sqrt{\frac{\delta(1-p)}{p}}. \tag{97}$$

Applying the simplex saturation logic from Proposition 3, the effective operational bounds are obtained by clamping the raw roots to $[0, 1/p]$:

$$\underline{r}_{f,\delta}(p) = \max\{\widetilde{\underline{r}}_{f,\delta}(p), 0\}, \qquad \overline{r}_{f,\delta}(p) = \min\left\{\widetilde{\overline{r}}_{f,\delta}(p), \frac{1}{p}\right\}. \tag{98}$$

Therefore, the final closed-form Band bounds are

$$\underline{r}_{\mathrm{TV},\delta}(p) = \max\left\{1 - \frac{\delta}{p}, 0\right\}, \qquad\qquad \overline{r}_{\mathrm{TV},\delta}(p) = \min\left\{1 + \frac{\delta}{p}, \frac{1}{p}\right\}, \qquad (99)$$

$$\underline{r}_{\chi^2,\delta}(p) = \max\left\{1 - \sqrt{\frac{\delta(1-p)}{p}}, 0\right\}, \qquad\qquad \overline{r}_{\chi^2,\delta}(p) = \min\left\{1 + \sqrt{\frac{\delta(1-p)}{p}}, \frac{1}{p}\right\}. \qquad (100)$$

$\square$

