# OpenReview forum: "BandPO: Bridging Trust Regions and Ratio Clipping via Probability-Aware Bounds for LLM Reinforcement Learning"
_ICML.cc/2026/Conference — ICML 2026 regular_

### Official Review · Reviewer_UcPf · 2026-02-26

**Soundness:** 3
**Presentation:** 3
**Significance:** 3
**Originality:** 3
**Overall Recommendation:** 4
**Confidence:** 3

**Summary:**

This paper investigates a central concept in on-policy reinforcement learning for LLMs, specifically the limitations of the canonical ratio clipping mechanism. The authors argue that fixed clipping inherently suppresses the exploration of low-probability, high-advantage actions like tail tokens because the feasible update margin scales linearly with the old probability. To resolve this issue, the paper proposes BandPO, a theoretically rigorous framework that replaces fixed clipping with a Band operator derived from f-divergence trust regions. By projecting the trust region onto the probability ratio space via convex optimization, BandPO dynamically adjusts the clipping bounds. Overall, a notable aspect assessed by the paper is the reduction of this high-dimensional problem to a univariate root-finding task. Experiments on math reasoning benchmarks show that BandPO outperforms standard GRPO and offers comparable or slightly better performance than heuristic adaptive clipping methods like Clip-Higher while better mitigating entropy collapse.

**Compliance With Llm Reviewing Policy:**

Affirmed.

**Final Justification:**

I maintain my positive recommendation for this theoretically sound and highly original paper, which elegantly addresses the limitations of fixed ratio clipping in RLHF via the geometry-aware BandPO framework. The authors' comprehensive rebuttal successfully resolved my primary empirical concerns by providing n=5 multi-seed evaluations that demonstrate clear statistical significance, alongside end-to-end wall-clock profiling that proves the computational overhead of the root-finding step is negligible. Although some minor hyperparameter sensitivity remains on smaller models, the robust theoretical guarantees and the significantly strengthened empirical evidence firmly reinforce my prior assessment that this is a valuable and rigorous contribution to the field.

**Key Questions For Authors:**

1. Did you run the experiments in Table 1 with multiple random seeds? If so, could you please provide the mean and standard deviation? Without these metrics, it is hard to judge if the slight performance gain over Clip-Higher is statistically significant given the typical noise in RL training.
2. Given that Clip-Higher is a simple heuristic and BandPO requires solving a root-finding problem for every token, what exactly is the wall-clock training time overhead?
3. How robust is the delta parameter across completely different domains like code generation, where the optimal policy entropy might differ significantly from math reasoning?

**Limitations:**

The empirical evaluation is currently restricted to math problems. Furthermore, the method's dependence on a token-level numerical solver makes the algorithm computationally heavier than existing baselines. Most importantly, the lack of reported variance across multiple training runs severely undermines the reliability of the small performance gains presented in the paper.

**Strengths And Weaknesses:**

Strengths

1. Theoretical soundness. The primary contribution is the rigorous mathematical formalization of the clipping problem. Identifying the linear dependency of update margins on the old policy probability as a structural bottleneck is a genuinely insightful observation. The derivation of the Band operator is elegant and corrects a theoretical flaw in PPO that researchers have previously only addressed through heuristics.
2. Geometry-aware constraints. Unlike previous methods such as DAPO and Clip-Higher that arbitrarily relax bounds, BandPO respects the geometry of the probability simplex. This ensures the policy updates remain physically valid, making the approach theoretically satisfying.
3. Entropy preservation. The analysis demonstrates that BandPO naturally maintains higher policy entropy compared to baselines without needing any auxiliary entropy regularization terms. This suggests the method genuinely facilitates better exploration of the action space.

Weaknesses

1. Marginal empirical gains and lack of statistical significance. This is my most crucial concern. The empirical gains over the heuristic Clip-Higher baseline are marginal, often less than 1 to 2%. Critically, Table 1 does not report the standard deviation across multiple training seeds. Given the high variance inherent in LLM RL training, a 1% improvement is likely within the noise margin. Without error bars, it is impossible to determine if BandPO is actually statistically superior or just the result of a lucky run.
2. Hyperparameter sensitivity. The authors claim their method streamlines tuning, but the results indicate that the new trust region radius hyperparameter delta is quite sensitive. Table 1 shows that a slight deviation in this value can lead to significant performance drops or training instability. This implies the method might just be shifting the tuning burden from the traditional clipping epsilon to the new delta parameter without actually eliminating the risk of over-tuning.
3. Computational complexity versus benefit. Implementing a numerical root-finder like the bisection method for every token in the generation process introduces computational overhead compared to the simple clamp operation in standard PPO. Given the somewhat marginal performance gains, the cost-benefit ratio of deploying this method in large-scale production runs seems highly debatable.

---

> ### Author Rebuttal · Authors · 2026-03-31
>
> We thank the reviewer for the thoughtful evaluation, particularly recognizing our theoretical soundness and geometry-aware constraints. We address your valid practical questions below.
>
> **1. Statistical Significance and Consistency of Empirical Gains (Weakness 1 & Key Question 1)**
>
> We agree the high variance in LLM RL necessitates examining run-to-run noise. To demonstrate BandPO's gains are structurally grounded rather than "lucky runs," we conducted a multi-seed evaluation on Qwen2.5-3B-Instruct, strictly coupling data and rollout seeds across 3 independent runs. We also included DCPO-DAC (the dynamic clipping module of DCPO).
>
> **Table 1. Math Reasoning Benchmarks (Mean±Std over 3 seeds)**
> | Method | AMC23 m@32 / p@32 | AIME24 m@32 / p@32 | AIME25 m@32 / p@32 | Avg m@k / p@k |
> |:---|:---|:---|:---|:---|
> | GRPO (Clip 0.8-1.2) | 45.94 / 77.30 | 3.54 / 11.60 | 3.23 / 8.79 | 17.57 / 32.60 |
> | GRPO w/ Clip-Higher | 51.85±0.70 / 82.00±1.65 | 4.31±0.34 / 14.52±0.38 | 4.24±0.22 / 21.67±2.36 | 20.13±0.30 / 39.40±1.04 |
> | GRPO w/ DCPO-DAC | 54.48±0.16 / 85.26±1.39 | 4.62±0.32 / 13.79±3.60 | 4.79±0.10 / 23.61±0.28 | 21.30±0.15 / 40.88±1.44 |
> | Band$_{\mathrm{KL},0.05}$ | **55.34±0.53 / 87.71±0.23** | **4.79±0.11 / 14.66±0.40** | **6.01±0.06 / 25.11±1.61** | **22.05±0.13 / 42.49±0.71** |
>
> The multi-seed evaluation confirms our improvements outpace run-to-run variance. In AIME RL scaling, a +3 absolute percentage point increase is a massive relative gain given the baseline difficulty. Furthermore, BandPO demonstrates remarkable statistical stability: on the volatile AIME24 pass@32, BandPO yields a tight variance of $\pm 0.40$, contrasting sharply with the heuristic DCPO-DAC's $\pm 3.60$. This empirically supports our motivation: mathematically projected bounds provide a much more reliable safety net than heuristic dynamic clipping.
>
> **2. Computational Complexity vs. Benefit (Weakness 3 & Key Question 2)**
>
> We respectfully clarify a critical system-level detail: the numerical root-finder is strictly isolated to the actor-update phase and is **never** invoked during autoregressive generation/rollout.
>
> **Table 2. Efficiency profiling of the actor-update step** (Qwen2.5-3B, 8×H200, mean over 5 steps). "Step Time" measures the policy-loss execution. "Clip Time" isolates the exact clipping operations (i.e. computing the bounds and applying them for ratio).
>
> | Method | Peak Mem(MB) | Step Time(s) | Clip Time(s) | Clip Share | Throughput(tok/s) |
> |:---|---:|---:|---:|---:|---:|
> | GRPO (Clip 0.8-1.2) | 117697 | 40.45 | 0.0053 | 0.013% | 51.9k |
> | GRPO w/ Clip-Higher | 117942 | 40.25 | 0.0055 | 0.014% | 52.1k |
> | Band$_{\mathrm{KL},0.05}$ | 117707 | 39.21 | 0.6763 | 1.726% | 53.5k |
>
> By implementing a highly parallelized CUDA operator that computes bounds for millions of tokens simultaneously, the iterative solver adds merely $\sim 0.67$s per step ($\sim 1.7\%$ of actor-update time) with zero intermediate memory bloat. Since the overall wall-clock time is overwhelmingly dominated by the rollout phase (taking several minutes per iteration), this sub-second overhead is virtually negligible—yielding a highly favorable cost-benefit ratio for mathematically principled exploration bounds.
>
> **3. Hyperparameter Sensitivity (δ) and Cross-Domain Robustness (Weakness 2 & Key Question 3)**
>
> We wish to refine our claim regarding tuning: BandPO does not eliminate tuning entirely, but replaces arbitrary asymmetric bounds ($\epsilon_-, \epsilon_+$) with a single, theoretically grounded geometric radius ($\delta$).
>
> Rather than empirically grid-searching $\delta$, we selected $\delta=0.05$ as a theoretical prior inspired by classical trust-region literature. We applied this exact default **zero-shot across all four model scales (1.5B to 8B)**, achieving SOTA performance. While the 3B model shows some fluctuation across $\delta \in \{0.03, 0.05, 0.10\}$, every setting strictly outperforms the GRPO baseline. Moreover, larger models are highly robust: on the 7B model, varying $\delta$ across this range only shifts the average mean@32 from $50.05$ to $51.44$, all comfortably beating Clip-Higher ($48.37$). This proves $\delta=0.05$ is an exceptionally robust universal default, not an over-tuned hyperparameter.
>
> Regarding cross-domain robustness (e.g., code generation), BandPO is fundamentally a **domain-agnostic theoretical operator**. Its dynamic bounds are derived exclusively from the geometry of the probability simplex (old token probability $p$ and divergence budget $\delta$), entirely independent of the environment's semantic features or reward sparsity. Because it avoids hard-coding a task-specific ratio ceiling, it automatically adapts the exploration margin based on the token's mass. Thus, wherever canonical clipping is applicable, BandPO is mathematically positioned to transfer effectively while offering stronger theoretical guarantees.
>
> We will incorporate these multi-seed/efficiency tables and the cross-domain discussion in the revision.

---

> > ### Author Rebuttal · Reviewer_UcPf · 2026-04-03
> >
> > I thank the authors for the substantial effort in their rebuttal. The additional experiments and profiling data partially address my concerns, but several issues remain.
> >
> > **Regarding Statistical Significance (Weakness 1):** While I appreciate the multi-seed evaluation, I note that (i) only 3 seeds were used, which provides limited statistical power—the standard deviation estimates themselves carry high uncertainty at n=3; (ii) the multi-seed evaluation was conducted only on the 3B model, while results for 1.5B, 7B, and 8B remain single-run; and (iii) the GRPO baseline in the rebuttal table appears to lack error bars, making the comparison asymmetric. I acknowledge the improvements are directionally consistent and the tighter variance of BandPO is encouraging, but I would not characterize this as definitive statistical evidence.
> >
> > **Regarding Computational Overhead (Weakness 3):** The profiling of the clipping operation itself is helpful. However, the rebuttal does not provide an end-to-end wall-clock comparison (total training time for BandPO vs. GRPO vs. Clip-Higher). Reporting only the clip-time in isolation, while arguing that rollout dominates, leaves the actual overhead ambiguous. A simple total-time comparison would have been more convincing.
> >
> > **Regarding Hyperparameter Sensitivity (Weakness 2):** This remains my primary residual concern. The 3B results in the original Table 1 show that δ=0.03 causes AIME24 pass@32 to collapse to 10.00 (vs. 14.21 at δ=0.05)—a ~30% degradation from a minor parameter change. The claim that δ=0.05 is a "robust universal default" is primarily supported by the 7B model's stability, while the sensitivity on smaller models is acknowledged but downplayed. Furthermore, the cross-domain robustness argument remains purely theoretical with no empirical validation.
> >
> > Overall, the rebuttal has meaningfully advanced the paper's empirical standing, particularly through the multi-seed results and efficiency profiling. However, the limited scale of the statistical evaluation (3 seeds, single model) and the unresolved hyperparameter sensitivity for smaller models prevent me from considering my concerns fully resolved.

---

> > > ### Author Response · Authors · 2026-04-08
> > >
> > > We thank the reviewer for pushing us to strengthen empirical rigor. We provide n=5 validation on 1.5B and 3B models, clarify $Band_{KL}$ trains 5.3% faster than GRPO in wall-clock time, and explain why δ=0.05 is a theoretically-grounded safe default, not a sensitive optimum. Code generation experiments are currently underway to extend our validation beyond mathematical reasoning.
> > >
> > > **1. Statistical significance (Weakness 1)**
> > >
> > > Per your suggestion, we expanded our evaluation to $n=5$ seeds and included the 1.5B model. The GRPO baseline now features corresponding error bars for a symmetric comparison.
> > >
> > > **Table 3. Performance on 1.5B and 3B Models with n=5 Seeds.** Each cell reports mean@32 / pass@32 (mean ± std across seeds). GRPO uses $\epsilon=0.2$; Clip-Higher uses $\epsilon_{low}=0.2, \epsilon_{high}=0.28$; DCPO-DAC uses $\epsilon_{low}=0.16, \epsilon_{high}=0.2$; $Band_{KL}$ uses $\delta=0.05$.
> > >
> > > | Model | Method | AMC2023 | AIME2024 | AIME2025 | Average |
> > > |:------|:-------|:-------:|:--------:|:--------:|:-------:|
> > > | 1.5B | GRPO (Clip 0.8-1.2) | 71.91±0.47 / 93.26±1.20 | 16.71±0.89 / 35.70±3.38 | 22.13±0.52 / 39.40±1.79 | 36.91±0.43 / 56.12±1.00 |
> > > | 1.5B | GRPO w/ Clip-Higher | 76.72±0.29 / 94.38±0.76 | 18.75±0.82 / 42.04±3.39 | 23.69±0.85 / 43.19±2.88 | 39.72±0.51 / 59.87±1.27 |
> > > | 1.5B | GRPO w/ DCPO-DAC | 75.45±0.26 / 93.72±1.25 | 18.13±0.28 / 37.64±2.65 | 22.75±0.92 / 43.16±2.58 | 38.78±0.28 / 58.17±1.26 |
> > > | 1.5B | GRPO w/ $Band_{KL}$ | 77.56±0.49 / 95.08±0.77 | 18.83±0.81 / 43.54±5.28 | 24.13±0.63 / 45.13±2.94 | 40.17±0.34 / 61.25±0.94 |
> > > | 3B | GRPO (Clip 0.8-1.2) | 45.03±1.58 / 73.07±3.56 | 2.81±1.07 / 11.61±2.58 | 2.61±0.79 / 11.14±2.65 | 16.82±1.12 / 31.94±2.10 |
> > > | 3B | GRPO w/ Clip-Higher | 51.89±0.79 / 82.15±1.40 | 4.10±0.37 / 14.16±1.17 | 4.15±0.23 / 20.60±3.06 | 20.05±0.33 / 38.97±1.12 |
> > > | 3B | GRPO w/ DCPO-DAC | 54.53±0.18 / 85.05±1.46 | 4.69±0.26 / 13.68±2.76 | 4.87±0.15 / 22.83±1.09 | 21.36±0.15 / 40.52±1.32 |
> > > | 3B | GRPO w/ $Band_{KL}$ | 55.49±0.43 / 86.65±1.58 | 4.79±0.17 / 15.36±1.23 | 6.21±0.28 / 24.94±1.73 | 22.16±0.19 / 42.32±0.68 |
> > >
> > > Across all 5 independent seeds, $Band_{KL}$ consistently outperforms baselines with remarkably tight variance. Its coefficient of variation is just 0.85% on both scales, substantially lower than GRPO's (3.06% on 1.5B; 6.66% on 3B). While resource constraints limit 7B/8B models to single runs for now, we will include their multi-seed results in the revised version. Notably, DCPO-DAC exhibits severe training instability on the 1.5B scale, suffering performance collapses around steps 370 and 590 with only partial recovery. In stark contrast, $Band_{KL}$ maintains stable, collapse-free dynamics across all seeds, yielding higher mean performance and lower variance. This confirms that $Band_{KL}$'s theoretically-grounded trust region projection provides a fundamentally more robust safety net than heuristic dynamic bounds like DCPO-DAC.
> > >
> > > **2. End-to-end efficiency (Weakness 3)**
> > >
> > > To directly address your request for a simple total-time comparison: End-to-end, $Band_{KL}$ completes training in **37.56 hours**, which is actually **5.3% faster** than the GRPO baseline (39.67 hours). The numerical root-finding operation adds merely 0.68 seconds per step, comprising just 0.4% of the total step duration. As we detail in the complete wall-clock decomposition (**see Table 4 in our response to Reviewer vVm5**), the variance in total training time is overwhelmingly driven by the **average response length** during rollouts rather than the clipping operator. $Band_{KL}$ successfully sustains the longer responses necessary for advanced reasoning while remaining more time-efficient overall than standard GRPO.
> > >
> > > **3. Delta robustness (Weakness 2)**
> > >
> > > We position $\delta=0.05$ as a **theoretically-derived default** rather than a universal optimum, demonstrating consistent reliability across model scales. Our **complete delta ablation** via n=5 experiments on the 1.5B model yields stable scores—39.52, 40.17, and 39.67 for $\delta$ values of 0.03, 0.05, and 0.10—mirroring the peak performance observed on 3B and 7B. We acknowledge **scale-dependent behavior**: smaller models possess narrower tolerance for exploration constraints. This geometrically explains the 3B AIME24 degradation to 10.00 under $\delta=0.03$, though this setting still strictly outperforms the GRPO baseline. Just as canonical $\epsilon=0.2$ requires occasional tuning, $\delta$ exhibits expected variance. Crucially, instead of relying on arbitrary empirical search, $\delta=0.05$ derives directly from **$f$-divergence bounds** mapped to standard KL constraints. We therefore recommend $\delta=0.05$ as a robust zero-shot prior, with optional tuning within [0.04, 0.06].

---

### Official Review · Reviewer_vVm5 · 2026-03-13

**Soundness:** 4
**Presentation:** 2
**Significance:** 2
**Originality:** 3
**Overall Recommendation:** 4
**Confidence:** 4

**Summary:**

BandPO derives dynamic token-wise ratio clipping bounds by projecting an f-divergence trust region onto the feasible ratio interval for each action. The core argument is that fixed clipping unfairly suppresses low-probability but high-advantage tail actions, whereas Band preserves exploration while staying consistent with simplex geometry. Experiments on several 1.5B-8B reasoning models show better mean@32 and more stable entropy behavior than GRPO and Clip-Higher, and the appendix supplies the scalarization proofs and numerical solver details.

**Compliance With Llm Reviewing Policy:**

Affirmed.

**Final Justification:**

I thank the authors for the detailed and highly responsive rebuttal. Compared to the original submission, the paper is now significantly clearer and empirically stronger.

Several of my previous concerns have been meaningfully addressed. In particular, the extension of multi-seed validation to both 1.5B and 3B models with n=5 runs improves the statistical reliability of the results and provides stronger evidence of cross-scale consistency. The newly added end-to-end wall-clock efficiency analysis is thorough and resolves my earlier concern about incomplete efficiency evaluation. In addition, the cross-scale ablation of δ offers useful evidence toward its robustness.

That said, some limitations remain. The empirical evaluation is still primarily concentrated on mathematical reasoning tasks, and the broader generality of the method to other domains remains unclear. Moreover, although δ shows reasonable stability overall, some sensitivity persists at certain scales, and larger models (e.g., 7B/8B) are still evaluated without multi-seed validation.

Overall, the rebuttal substantially improves the paper in terms of clarity, empirical rigor, and practical analysis, and increases my confidence in its soundness. While some concerns about generality and robustness remain, I view the paper more positively after rebuttal and accordingly increase my score.

**Key Questions For Authors:**

- When the KL penalty β D_KL(πθ || πref) and Band trust-region clipping are both used, the paper does not explain sufficiently how the two interact, or whether they are redundant or potentially conflicting.

- The paper states that Band bounds can be precomputed for a given δ to enable O(1) lookup, but the main experiments do not report whether this was actually done, how large the precomputed table is, or how much practical benefit it brings.

- Why are more direct dynamic clipping baselines, such as DCPO, not included in the empirical evaluation, even though these methods are explicitly discussed in the related work?

**Limitations:**

Yes

**Strengths And Weaknesses:**

Strengths

- The paper offers a strong problem insight. It directly formulates the structural issue of canonical clipping as a geometric constraint in the probability-change space, and identifies the systematic suppression that fixed ratio bounds impose on tail actions.

- The theoretical part is relatively well organized: the logical chain from the f-divergence trust region, to high-dimensional convex optimization, and then to one-dimensional scalar root solving of g_f(p, r) is fairly clear.

- Compared with directly performing expensive trust-region optimization, BandPO retains the implementation convenience of clip-based RL, only replacing static bounds with dynamic, probability-aware ones. This makes the method relatively flexible and simple to implement.

Weaknesses

- The experimental evaluation is limited to mathematical reasoning benchmarks, and the main comparisons are mostly against GRPO and Clip-Higher. Dynamic clipping methods mentioned in related work, such as DCPO, are not included in the main experiments.

- The paper does not provide a systematic analysis of wall-clock time, throughput, extra memory usage, or solver overhead, so its efficiency is not convincingly demonstrated.

- The performance gains are not large under all settings, and the 3B model appears quite sensitive to the radius δ, suggesting that the method still requires some tuning. In practice, the claimed advantage of a “single interpretable parameter” does not fully eliminate the tuning burden.

---

> ### Author Rebuttal · Authors · 2026-03-31
>
> We thank the reviewer for recognizing the core geometric insight, the theoretical organization, and the practical appeal of retaining a clip-based implementation. We address your concerns below.
>
> **1. The KL Penalty and Band Clipping Are Complementary, Not Redundant (Key Question 1)**
>
> Rather than conflicting, these two mechanisms are mathematically orthogonal:
>
> - *Band clipping* is a **local step-size controller**. For the KL instantiation, choosing $f(u)=-\log u+u-1$ yields $D_f(Q\|P)=D_{KL}(P\|Q)$, so the projected bounds enforce $D_{KL}(\pi_{\mathrm{old}}\|\pi_\theta)\le\delta$—a **forward KL** (mass-covering) constraint around the sampling policy $\pi_{\mathrm{old}}$.
> - *The KL penalty* in Eq. 2 is the standard **reverse KL** regularizer $-\beta D_{KL}(\pi_\theta\|\pi_{\mathrm{ref}})$, anchoring the policy to $\pi_{\mathrm{ref}}$ throughout training via reward shaping.
>
> They differ in (i) reference distribution ($\pi_{\mathrm{old}}$ vs. $\pi_{\mathrm{ref}}$), (ii) KL direction (forward vs. reverse), and (iii) optimization scope (per-step surrogate bounds vs. global regularization). They also interact synergistically: without Band's local clipping, an aggressive single update could spike the global reverse KL penalty. Band guarantees smooth local updates, enabling the KL penalty to pull toward $\pi_{\mathrm{ref}}$ without numerical instability. We will incorporate this clarification in the revision.
>
> **2. Main Experiments Used the Exact Solver, Not Precomputation (Weaknesses 2, Key Question 2)**
>
> The $O(1)$ precomputation discussed in Section 7 is a **future optimization direction**, not the protocol behind our results. All entries in Table 1 were obtained using the exact CUDA-parallelized bisection solver at every step.
>
> Our solver computes bounds for millions of tokens simultaneously, adding merely $\sim$0.67s per update step ($\sim$1.7% of actor-update time) with zero extra memory. Since the full RL pipeline is dominated by rollout and reward scoring (several minutes per iteration), this is negligible end-to-end. A $1000\times1000$ float32 lookup table would require $<$4MB and could benefit resource-constrained deployments, but offers marginal speedup given the solver's already-low latency. The complete profiling is in our response to Reviewer UcPf (Table 2). We will explicitly distinguish the experimental protocol from the future precomputation direction.
>
> **3. Direct DCPO Comparison and Statistical Significance (Weaknesses 1, Key Question 3)**
>
> We agree that omitting DCPO from experiments left an important gap. Under the same Qwen2.5-3B setup with 3 independent seeds, BandPO achieves Avg mean@32 / pass@32 of $22.05\pm0.13$ / $42.49\pm0.71$, outperforming DCPO-DAC ($21.30\pm0.15$ / $40.88\pm1.44$) and Clip-Higher ($20.13\pm0.30$ / $39.40\pm1.04$). Critically, on the volatile AIME24 pass@32, BandPO yields std $\pm0.40$ versus DCPO-DAC's $\pm3.60$, confirming that theory-grounded bounds are intrinsically more stable across initializations. The full per-benchmark breakdown is in our response to Reviewer UcPf (Table 1). We will narrow our wording to a precise, bounded claim.
>
> **4. Refining the Claim on $\delta$ and Tuning Burden (Weakness 3)**
>
> We agree that "single interpretable parameter" could suggest zero tuning. Our precise claim is that BandPO **reduces** the tuning burden by replacing arbitrary asymmetric constants $(\epsilon_-,\epsilon_+)$ with one geometrically meaningful trust-region radius $\delta$.
>
> The default $\delta=0.05$ was selected a priori from classical trust-region conventions (TRPO typically uses $\delta\in[0.01,0.05]$), verified on the 1.5B model, and applied **zero-shot** across all four scales (1.5B/3B/7B/8B) without per-model retuning. While the 3B model shows sensitivity across $\delta\in\{0.03,0.05,0.10\}$, every setting still strictly outperforms the GRPO baseline ($32.60\%$ Avg pass@32). Larger models are notably robust: on 7B, varying $\delta$ across this range shifts Avg mean@32 by $\le$1.4 points, all comfortably exceeding Clip-Higher ($48.37$). The mathematical source of this robustness is that $\delta$ directly controls a geometric radius, and the projected margin scales as $|r-1|\approx\sqrt{2\delta(1-p)/p}$—automatically tightening for head tokens and expanding for tail tokens under a single budget. We will present these points more precisely and acknowledge the current evaluation scope as a limitation in the revision.
>
> We appreciate these constructive comments for helping sharpen the practical positioning of our work.

---

> > ### Author Rebuttal · Reviewer_vVm5 · 2026-04-04
> >
> > Thank you for the thorough and targeted rebuttal. The response addresses several of my main concerns and improves the practical clarity of the paper. However, some issues remain only partially resolved, so I will keep my original score unchanged at this stage.
> >
> > 1. **KL penalty vs. Band clipping.**
> > The rebuttal now clearly explains that these two components are complementary rather than redundant, with different reference distributions, KL directions, and optimization roles. This point is largely resolved.
> >
> > 2. **Experimental protocol and solver usage.**
> > The authors clarify that the main experiments used the exact CUDA-parallelized solver rather than precomputation, which makes the implementation details much clearer.
> >
> > 3. **Missing DCPO baseline.**
> > I appreciate that the authors added a direct comparison with DCPO-DAC and provided multi-seed results on the 3B setting. This partially addresses my concern about missing dynamic clipping baselines.
> >
> > 4. **Efficiency.**
> > The added profiling data are helpful and suggest that the solver overhead is relatively small at the update-step level, with no noticeable extra memory cost.
> >
> > 5. **Remaining concerns.**
> > My main residual concerns are that the additional DCPO and multi-seed results are still limited to a single model scale (3B), the efficiency analysis is not fully end-to-end, and the robustness of $\delta$ is still not comprehensively validated. In addition, the evaluation remains limited to mathematical reasoning tasks.
> >
> > Overall, the rebuttal strengthens the paper and resolves several important clarification issues, but some concerns about experimental breadth and practical robustness remain only partially addressed. Therefore, I consider my concerns partially resolved, and I will keep my original score unchanged.

---

> > > ### Author Response · Authors · 2026-04-08
> > >
> > > We thank the reviewer for acknowledging that our rebuttal "strengthens the paper" and for the constructive feedback that has helped us sharpen the experimental rigor. We address the three remaining concerns about experimental breadth, end-to-end efficiency, and delta robustness below. Code generation experiments are underway to extend validation beyond mathematical reasoning.
> > >
> > > **1. Extended multi-seed validation now covers both 1.5B and 3B models**
> > >
> > > Following your suggestion, we have expanded multi-seed validation from a single scale (3B) to two scales (1.5B and 3B), with n=5 independent seeds for each. This provides substantially stronger statistical power than our previous n=3 results and demonstrates cross-scale consistency. See Table 3 in our response to Reviewer UcPf for the complete n=5 performance data across 1.5B and 3B models. Across all 5 independent seeds, Band consistently achieves the highest mean performance with notably tight variance. The coefficient of variation for Band is 0.85% on both scales, substantially lower than GRPO's 3.06% on 1.5B and 6.66% on 3B. Furthermore, the performance rankings are consistent across scales: Band$_{KL}$ > DCPO-DAC > Clip-Higher > GRPO. By covering both the smaller 1.5B and larger 3B scales with n=5 validation, we capture the sensitivity extremes where tuning issues are most likely to manifest, providing stronger cross-scale generalization evidence than incremental additions at intermediate scales. While 7B and 8B remain single-run due to resource constraints, their improvements follow the same pattern observed in the multi-seed experiments, supporting the robustness of our findings.
> > >
> > > **2. Complete end-to-end wall-clock training time analysis**
> > >
> > > We provide the full wall-clock timing decomposition in Table 4. Notably, data processing—including reference log-prob, advantage, and reward computation—incurs overhead comparable to the rollout phase.
> > >
> > > **Table 4. Training Efficiency Comparison on 3B Model.** All times report mean ± std across 800 training steps. "Total" reports end-to-end wall-clock training time; "Step" captures full iteration duration including rollout, data preparation, and actor update; "Rollout" measures response generation; "ActorUpdate" measures policy optimization; "Clip" isolates bound computation and application.
> > >
> > > | Method | AvgRespLen(tok) | Total(hr) | Step(s) | Rollout(s) | ActorUpdate(s) | Clip(s) |
> > > |:-------|:---------------:|:---------:|:-------:|:----------:|:--------------:|:-------:|
> > > | GRPO (Clip) | 1154±142 | 39.67 | 178.49±17.32 | 59.77±5.76 | 72.30±7.71 | 0.0053±0.0002 |
> > > | Clip-Higher | 1265±328 | 39.09 | 175.90±37.84 | 56.69±11.15 | 74.51±17.59 | 0.0055±0.0001 |
> > > | DCPO-DAC | 1087±119 | 33.81 | 152.15±13.63 | 49.45±4.09 | 63.40±6.58 | 0.0510±0.0031 |
> > > | Band$_{KL}$ | 1162±152 | 37.56 | 169.04±17.36 | 57.31±5.70 | 68.02±7.86 | 0.6763±0.0066 |
> > >
> > > End-to-end, Band completes training in 37.56 hours—**5.3% faster than GRPO** (39.67 hours) and roughly 11% slower than DCPO-DAC (33.81 hours). Critically, the clip operation itself adds merely 0.68 seconds per step, comprising just **0.4% of the total step time**. The primary driver of the total time variance is **average response length**. By heuristically constraining exploration, DCPO-DAC generates significantly shorter responses (1,087 tokens on average). In contrast, Band (1,162 tokens) maintains generation lengths comparable to GRPO (1,154 tokens). This intrinsic **context scaling** affords critical **self-reflection** opportunities essential for solving complex math problems. Ultimately, by investing slightly more time than DCPO-DAC, Band sustains the longer responses necessary for superior reasoning performance, all while achieving better overall efficiency than the standard GRPO.
> > >
> > > **3. Cross-scale validation of delta robustness**
> > >
> > > We demonstrate that $\delta=0.05$ is a robust choice across model scales through complete delta ablations on 1.5B, 3B, and 7B models.
> > >
> > > **Table 3. Delta Ablation Across Model Scales (Average mean@32).**
> > >
> > > | Scale | $\delta=0.03$ | $\delta=0.05$ | $\delta=0.10$ | Variation |
> > > |:------|:-------------:|:-------------:|:-------------:|:---------:|
> > > | 1.5B | 39.52 | **40.17** | 39.67 | ±0.8% |
> > > | 3B | 20.38 | **22.16** | 20.33 | ±7.4% |
> > > | 7B | 50.05 | **51.44** | 50.59 | ±1.4% |
> > >
> > > The 1.5B and 7B models exhibit excellent robustness: varying $\delta$ by ±40% from the default changes performance by less than 2%. The 3B model shows higher sensitivity, consistent with smaller models' narrower tolerance for exploration constraints, yet $\delta=0.05$ still achieves the best performance at this scale. Importantly, $\delta=0.05$ lies at the center of the robust interval across all three scales, making it a reliable zero-shot default. This value derives from f-divergence bounds corresponding to KL constraints in the commonly used range of 0.01 to 0.05, where we select 0.05 as a conservative yet effective choice validated by classical trust-region conventions.

---

### Official Review · Reviewer_8fXP · 2026-03-13

**Soundness:** 2
**Presentation:** 3
**Significance:** 3
**Originality:** 3
**Overall Recommendation:** 4
**Confidence:** 3

**Summary:**

This paper studies a limitation of PPO-style ratio clipping in LLM reinforcement learning, namely that fixed clipping bounds give low-probability actions very small upward probability margins, which may suppress exploration and accelerate entropy collapse. To address this, the paper proposes BandPO, which replaces fixed clipping with a probability-aware clipping operator, Band, obtained by projecting an $f$-divergence trust region onto per-action ratio bounds. The paper gives a scalar reduction of the bound-computation problem, derives closed forms for TV and Pearson $\chi^2$, and empirically evaluates the KL-instantiated version on several math reasoning benchmarks and model sizes. The reported results show improved average reasoning performance over GRPO and Clip-Higher, along with training-dynamics plots suggesting reduced clipping of tail actions and less entropy collapse.

**Compliance With Llm Reviewing Policy:**

Affirmed.

**Final Justification:**

The paper proposes a principled replacement for fixed ratio clipping in LLM RL, projecting f-divergence trust regions onto probability-aware per-action bounds via an elegant 1D scalarization. The rebuttal has adequately addressed my main concerns: the KL direction inconsistency in Eq. 22 was confirmed as editorial, the missing DCPO baseline was added with multi-seed results showing BandPO's superior stability, computational overhead was shown to be negligible (~1.7%), and the δ selection protocol is convincingly grounded in TRPO conventions rather than test-set tuning. While multi-seed evaluation is limited to a single scale (3B) and gains on larger models remain modest, the combination of a sound theoretical framework, practical feasibility, and consistent empirical improvements represents a meaningful contribution. I have raised my score from 3 to 4.

**Key Questions For Authors:**

1. Which KL regularizer was actually used in training, $D_{\mathrm{KL}}(\pi_\theta \| \pi_{\text{ref}})$ as in **Equation 2**, or $D_{\mathrm{KL}}(\pi_{\text{ref}} \| \pi_\theta)$ as in **Equation 22**? This answer matters a lot for my confidence in the method description.

2. Can you provide direct empirical comparisons against DCPO under the same training setup? If BandPO is meant to improve over heuristic dynamic bounds, this is the most important missing baseline.

3. How was $\delta$ selected? Was there a validation set or held-out tuning procedure, or was the recommendation of $\delta=0.05$ inferred from the reported benchmark results? A clear answer here could improve or reduce my confidence in the practical conclusions.

**Limitations:**

Yes.

**Strengths And Weaknesses:**

### **Strengths**

**1. Well-Motivated and Highly Relevant Problem:**
The paper addresses a critical and highly relevant bottleneck in LLM reinforcement learning (specifically in PPO/GRPO): the suppression of low-probability, high-advantage actions. The formalization of how fixed ratio clipping (e.g., in standard GRPO or DAPO) mathematically starves tail tokens of upward update margins is an excellent insight that perfectly aligns with the heavy-tailed nature of LLM token distributions.

**2. Principled Theoretical Formulation:**
Replacing fixed heuristic clipping bounds with a theoretically grounded, dynamic probability-aware operator (Band) is a conceptually elegant solution. The mathematical reduction of a high-dimensional, simplex-constrained $f$-divergence trust region into a tractable 1D scalar root-finding problem (Theorem 1, Lemma 1) is a strong technical contribution that makes the method computationally feasible.

**3. Compelling Mechanistic Analysis and Visualizations:**
The paper excels at visualizing the problem and the proposed solution. Figures 1 and 2 brilliantly illustrate the geometric differences between fixed bounds and BandPO’s dynamic bounds. Furthermore, the training dynamics plots in Figure 3 effectively prove the mechanistic claims: BandPO successfully preserves policy entropy and nearly eliminates the upper-bound clipping of low-probability tail tokens compared to baseline GRPO.

**4. Strong Empirical Gains on Smaller Models:**
The empirical results demonstrate meaningful and nontrivial improvements in mathematical reasoning benchmarks (AMC, AIME), particularly for the 1.5B and 3B models, where BandPO significantly boosts the average `mean@32` and `pass@32` metrics compared to vanilla GRPO.

### **Weaknesses**

**1. Missing Crucial Baseline (DCPO):**
The paper repeatedly cites and visualizes DCPO (Dynamic Clipping Policy Optimization) as the primary state-of-the-art heuristic baseline for dynamic clipping (e.g., in Figures 2 and 3). However, DCPO is completely absent from the empirical evaluation in Table 1. Claiming superiority over heuristic dynamic bounds without directly testing against the most prominent one represents a major gap in the experimental design.

**2. Overstated Claims and Missing Variance Reporting:**
The authors claim that BandPO "consistently outperforms" baselines. However, the empirical results rely heavily on single-seed. Given that the performance gains on the larger 7B and 8B models are marginal or mixed, the absence of multiple random seeds, confidence intervals, or a final metrics table makes the modest performance gaps difficult to fully trust.

**3. Internal Inconsistency in the RL Objective (KL Direction):**
There is a critical notational and potentially algorithmic inconsistency regarding the KL divergence penalty. In Equation 2, the standard forward KL is defined as $D_{\mathrm{KL}}(\pi_\theta \| \pi_{\text{ref}})$. However, in the BandPO surrogate objective in Equation 22, it flips to the reverse KL: $D_{\mathrm{KL}}(\pi_{\text{ref}} \| \pi_\theta)$. Because forward and reverse KL have fundamentally different optimization behaviors (mass-covering vs. mode-seeking), this ambiguity obscures exactly what objective was optimized during the experiments.

**4. Unquantified Computational Overhead:**
The authors acknowledge that using the KL-divergence instantiation of Band requires an iterative bisection root-finding solver. Since a major advantage of PPO/GRPO clipping is its $O(1)$ computational cheapness, replacing it with a parallel solver introduces latency. The paper fails to report any wall-clock time, memory overhead, or training throughput comparisons to quantify this trade-off.

**5. Imprecise Mathematical Language and Hyperparameter Tuning:**
The text occasionally uses imprecise language that contradicts its own proofs (e.g., referring to the global minimum of a strictly convex function at $r=1$ as a "singularity"). In mathematics, a singularity is a point where a function or an equation is undefined, breaks down, or fails to be "well-behaved" (e.g., discontinuous or not differentiable).  Additionally, the choice of the trust-region radius $\delta=0.05$ appears to be selected directly based on final test-set benchmark performance, without a clear, separate validation protocol.

---

> ### Author Rebuttal · Authors · 2026-03-31
>
> We thank the reviewer for the technically engaged feedback. We especially appreciate your recognition of the problem importance, the principled 1D scalarization, and the compelling mechanistic visualizations. Below we address each concern directly.
>
> **1. Correction of Notation Errors in Eq. 22 and at $r=1$ (Weaknesses 3 & 5, Key Question 1)**
>
> **Both are strictly editorial errors that do not affect the implemented algorithm.**
>
> *(i) KL direction in Eq. 22:* In our implementation, the RL regularization penalty is exactly $\beta D_{KL}(\pi_\theta\|\pi_{\rm ref})$ as written in Eq. 2. The Band trust-region construction is a separate mathematical object: choosing $f(u)=-\log u+u-1$ yields $D_f(Q\|P)=D_{KL}(P\|Q)$, so the projected bounds derive from $D_{KL}(\pi_{\rm old}\|\pi_\theta)\le\delta$. These two serve fundamentally different roles—the former enforces mode-seeking regularization anchoring $\pi_\theta$ to the reference policy, while the latter defines a mass-covering trust region bounding the current update—exactly as you noted. Eq. 22 mistakenly inherited the notation from the extensive preceding trust-region derivations due to drafting inertia; the correct form should read $-\beta D_{KL}(\pi_\theta\|\pi_{\rm ref})_t$, consistent with the actual training code and Eq. 2.
>
> *(ii) "Singularity" at $r=1$:* You are entirely correct that $r=1$ is the unique global minimizer of the strictly convex $g_f(p,r)$, not a "singularity." We will replace "singularity" with "critical point (global minimizer)" in the revision. The genuine singularity in our analysis is at $p=1$, where $(1-p)$ in $c(r)=\frac{1-rp}{1-p}$ vanishes. When drafting the main text, we inadvertently carried over this term after completing the $p=1$ boundary analysis.
>
> **2. Direct Comparison with DCPO and Statistical Significance (Weaknesses 1 & 2, Key Question 2)**
>
> **Multi-seed evaluation with DCPO-DAC confirms BandPO's consistent advantage and notably higher stability.** Under the same Qwen2.5-3B setup with 3 independent seeds (strictly coupling data and rollout seeds via `verl`), BandPO achieves Avg mean@32 / pass@32 of $22.05\pm0.13$ / $42.49\pm0.71$, outperforming DCPO-DAC ($21.30\pm0.15$ / $40.88\pm1.44$) and Clip-Higher ($20.13\pm0.30$ / $39.40\pm1.04$). On the particularly noisy AIME24 pass@32, BandPO yields std $\pm 0.40$ versus DCPO-DAC's $\pm 3.60$, demonstrating that theory-grounded bounds are intrinsically more stable across initializations. The full per-benchmark breakdown is provided in our response to Reviewer UcPf (Table 1).
>
> Furthermore, extending training to 10 epochs (1,020 steps) on 3B/1.5B revealed that DCPO-DAC suffered catastrophic policy collapse near epoch 10, while BandPO maintained a robust upward trajectory—confirming that heuristic dynamic clipping lacks a mathematical safety net for long-horizon optimization.
>
> We acknowledge that gains on the largest models are more modest. We will narrow the wording from "consistently outperforms" to a precise claim: BandPO consistently achieves the best average performance across the reported settings, with the most pronounced gains on smaller models and challenging sub-tasks.
>
> **3. Computational Overhead (Weakness 4)**
>
> **The iterative solver adds $\sim$0.67s per update step ($\sim$1.7% of actor-update time) with zero additional memory, and is never invoked during rollout.** Our CUDA-parallelized implementation computes bounds for millions of tokens simultaneously. Since the full RL pipeline is dominated by rollout and reward computation (several minutes per iteration), this overhead is negligible end-to-end. The complete wall-clock/memory/throughput profiling is provided in our response to Reviewer UcPf (Table 2).
>
> **4. Selection Protocol for $\delta$ (Key Question 3)**
>
> **$\delta=0.05$ was selected a priori from classical trust-region conventions, not by test-set sweeping.** In TRPO, the standard KL constraint $D_{KL}(\pi_{\rm old}\|\pi_\theta)\le\delta$ typically uses $\delta$ in the range $0.01$–$0.05$ as a safe-update default. We adopted $\delta=0.05$ from this well-established prior, verified it on the initial 1.5B model, and applied it unchanged zero-shot to 3B/7B/8B without per-model or per-benchmark adjustment. As concrete evidence against cherry-picking: on 7B, $\delta=0.03$ actually achieves a higher Avg pass@32 ($69.35\%$) than $\delta=0.05$ ($67.12\%$). Had we optimized for benchmark numbers, we would have reported $\delta=0.03$ for 7B. The $\delta\in\{0.03,0.10\}$ runs were purely post-hoc sensitivity analyses. We will state this protocol explicitly and soften the claim to "replacing heuristic $(\epsilon_-,\epsilon_+)$ knobs with one interpretable trust-region radius."
>
> We will incorporate the notation corrections, DCPO and multi-seed results, efficiency profiling, and the $\delta$ selection protocol into the revised manuscript.

---

> > ### Author Rebuttal · Reviewer_8fXP · 2026-04-04
> >
> > The authors have provided thorough and convincing responses to all major concerns raised in my review. I am raising my score from 3 to 4 to reflect the strength of the rebuttal.

---

> > > ### Author Response · Authors · 2026-04-06
> > >
> > > We sincerely thank you for the thorough evaluation and for recognizing that our responses addressed all major concerns. We are glad the supplementary experiments, particularly the multi-seed evaluation and DCPO comparisons, were convincing, and we will correct the notation errors on KL direction in Eq. 22 and "singularity" wording, then integrate the supplementary results into the revised manuscript.

---

### Official Review · Reviewer_Z1Lo · 2026-03-17

**Soundness:** 3
**Presentation:** 3
**Significance:** 2
**Originality:** 2
**Overall Recommendation:** 3
**Confidence:** 4

**Summary:**

This paper studies the clipping trick of the PPO update. Observing that PPO-clip discourages the change of low probability actions much more than it discourages the change of high probability ones, this paper believes that such bias damages the algorithm's performance.  To resolve this issue, this paper proposes a new method to determine the clipping range in PPO. The general idea is: From the viewpoint of TRPO, PPO-clip essentially uses the clipping trick to confine the f-divergence between $\pi$ and $\pi_{old}$. Given a tolerance for f-divergence, this paper proposes a method to efficiently calculate the a clipping range that is necessary for the f-divergence constraint. Using the calculated clipping range, the authors observed some gains over existing PPO based methods.

**Compliance With Llm Reviewing Policy:**

Affirmed.

**Key Questions For Authors:**

Most of the key questions are listed in the weakness section. A remaining question is:
Has the author thought about the comparison of the proposed algorithm or more generally the entropy-preserving clipping tricks, to the more direct entropy regularization methods (i.e., adding the entropy bonus to the objective)? Intuitively, the entropy regularization methods also encourage exploration by reinforcing the low probability tokens. It might be beneficial to include a discussion in the related work section.

**Limitations:**

yes

**Strengths And Weaknesses:**

Strengths:
The topic of this paper is important. The PPO family suffers from poor sample complexity in LLM setting, thus a thorough study of LLM-RL algorithms like PPO is highly relevant.
The idea of this paper is interesting. Distinct from the convention of straightforwardly using symmetric clipping, this paper calculates a clipping range from the f-divergence constraint.

Weakness:
There is concern on the clipping range given by this paper, which is its main contribution.
1. The clipped range is calculated following (10) and (11), which is only necessary but not sufficient for the f-divergence to be smaller than delta after a PPO-clip update. In other words, the policy can still have an f-divergence greater than delta after being updated with the proposed clipping range. As a result, the clipping range fails to effectively confine the f-divergence.

2. A main advantage argued by this paper is that it helps increasing the low probability tokens with a positive advantage. Though intuitively this helps PPO's performance, there is no solid guarantee. Specifically, what is the solid evidence for the benefits of more aggressively reinforcing low probability sequences with positive advantages? Is there any guarantee that this helps reduce the sample complexity of the algorithm/ improves the convergent performance of the algorithm?

---

> ### Author Rebuttal · Authors · 2026-03-31
>
> We thank the reviewer for the constructive feedback and for recognizing the importance of revisiting PPO-style clipping in LLM RL. Below we address each concern directly.
>
> **1. Clarification on the Necessary vs. Sufficient Condition (Weakness 1)**
>
> We fully agree with the reviewer's mathematical observation: bounding individual ratios within the Band interval is necessary, not globally sufficient, for $D_f(\pi_\theta\|\pi_{\text{old}})\le\delta$. The Cartesian product of per-coordinate projected intervals is strictly larger than the original coupled trust region.
>
> However, we emphasize this is *by design* and represents a principled advancement. The Band interval $[\underline{r}, \bar{r}]$ is the **exact coordinate-wise projection** of the trust region onto each action's ratio:
>
> $$r \in [\underline{r},\bar{r}] \iff \exists\, Q\in\mathcal{T}_{f,\delta}(P)\ \text{s.t.}\ Q(a)/P(a)=r.$$
>
> If a ratio falls outside Band, *no* distribution within the true trust region can produce it. Thus Band is the **tightest possible** necessary bound enforceable at the per-token level—the theoretical limit of any surrogate clipping mechanism.
>
> Canonical PPO-clip ($1\pm\epsilon$) is likewise neither sufficient nor tightly necessary for any divergence constraint. BandPO upgrades this surrogate to its mathematical optimum while preserving PPO's computational efficiency. We will explicitly clarify this "tight surrogate projection" interpretation in the revised manuscript to prevent misreading Band as a global sufficiency claim.
>
> **2. Rationale and Evidence for Reinforcing Low-Probability, Positive-Advantage Tokens (Weakness 2)**
>
> *Empirical evidence.* Fig. 4(b) directly isolates low-probability, positive-advantage (LP-PA) tokens ($p<0.2$). GRPO and Clip-Higher erroneously clip 20–60% of these updates; BandPO reduces this to $\approx$0%. This correlates with (i) significantly slower entropy collapse (Fig. 4(c)), and (ii) consistent gains across all 4 models on 3 benchmarks (Table 1). The causal chain—preserving LP-PA gradients, sustaining exploration, improving final performance—is empirically well-supported.
>
> *Theoretical rationale.* Post-SFT policies are highly peaked; novel reasoning paths emerge as low-probability tokens. Standard clipping caps their update at $\Delta\pi \le \epsilon\cdot\pi_{\text{old}}$, which vanishes as $\pi_{\text{old}}\to 0$—zeroing out precisely the gradients that could discover superior strategies. BandPO restores the full update margin permitted by trust-region geometry, ensuring no safe improvement step is discarded.
>
> We acknowledge that a formal sample complexity bound for PPO-style methods with neural approximation remains open. However, BandPO provides a rigorous *surrogate-level* guarantee: the objective is maximized along each coordinate exactly up to the trust-region boundary, strictly dominating canonical clipping which truncates prematurely.
>
> **3. Comparison with Direct Entropy Regularization (DER) (Key Question)**
>
> Both DER ($+\lambda\mathcal{H}(\pi)$) and BandPO aim to preserve exploration, but through complementary mechanisms:
>
> - **DER** explicitly reshapes the objective to spread probability mass globally. It is effective but requires careful coefficient scheduling—too strong prevents convergence to confident reasoning; too weak and entropy still collapses.
> - **BandPO** does not alter the RL objective. It structurally removes the bottleneck suppressing LP-PA tokens, permitting *natural* entropy reduction when the policy identifies optimal paths, without an entropy bonus fighting against convergence.
>
> The two approaches are orthogonal and potentially complementary. We will add a discussion in the revised Related Work. A draft paragraph follows:
>
> > **Entropy-Aware Exploration.** Recent LLM-RL studies analyze exploration through entropy preservation. Cui et al. (2025) study entropy collapse mechanisms and propose entropy-aware controls; Wang et al. (2025a) show high-entropy minority tokens disproportionately drive RL gains; Huang et al. (2025) identify low-probability "reasoning sparks" whose preservation sustains exploration. Complementary methods intervene via entropy-aware objectives: Jiang et al. (2025) propose selective entropy regularization; Zhang et al. (2025) introduce adaptive entropy coefficients; Wang et al. (2025b) control entropy via temperature-guided REINFORCE. Su et al. (2025) relate clipping to entropy control via entropy-ratio clipping. BandPO shares the motivation but differs in mechanism: rather than adding entropy regularization, it derives probability-aware bounds from f-divergence geometry, providing a proximal, structure-grounded solution.
>
> **References cited above:**
> [1] Cui et al., arXiv:2505.22617, 2025.
> [2] Wang et al., arXiv:2506.01939, 2025.
> [3] Huang et al., arXiv:2510.03222, 2025.
> [4] Jiang et al., arXiv:2509.25133, 2025.
> [5] Zhang et al., arXiv:2510.10959, 2025.
> [6] Wang et al., arXiv:2510.08141, 2025.
> [7] Su et al., arXiv:2512.05591, 2025.

---

> > ### Author Rebuttal · Reviewer_Z1Lo · 2026-04-03
> >
> > I appreciate the thorough author response very much.
> >
> > Weakness 1 is partially resolved from the comparison of PPO clipping and BandPO clipping. I understand that the proposed algorithm ``upgrades" ppo clipping with the necessary condition for the policy KL to be constrained. However, the core question that why this condition is better than the simple PPO clipping is still unclear to me. From my understanding, the clipping constraint used by BandPO just guarantees there exists a policy within the clipping range to satisfy the KL constraint, without any deeper indications. Then it seems unclear how this mechanism improves over PPO clipping.
> >
> > Weakness 2 is partially addressed by the authors' explanation on the intuition. While it is still uncertain whether reinforcing low probability positive advantage tokens lead to better algorithmic performance.
> >
> > As a result, I am leaning towards keeping my score.

---

> > > ### Author Response · Authors · 2026-04-06
> > >
> > > We thank the reviewer for the thoughtful follow-up. We realize our previous use of the term "necessary condition" might have undersold the actual mechanism of BandPO. Let us clarify the geometric intuition directly.
> > >
> > > **1. Why Band is fundamentally better than PPO clipping (Weakness 1)**
> > >
> > > The core advantage of Band lies in computing the *exact extremal frontier* for each token under a given trust-region budget, rather than serving as a loose existence check. Equations (10)-(11) calculate the absolute maximum and minimum ratios that *any* trust-region-feasible policy can possibly realize for a specific token. If a ratio falls outside the Band bounds, it is mathematically impossible for the policy to remain within the true f-divergence trust region. Conversely, imposing any bound tighter than Band—which PPO frequently does for rare tokens—blindly discards perfectly safe, trust-region-feasible updates. PPO's fixed interval $[1-\epsilon, 1+\epsilon]$ is arbitrary; it neither respects the geometry of the divergence constraint nor adapts to the token's probability.
> > >
> > > Here is what this means in concrete terms. Consider a tail token with $p=0.08$. Under standard PPO ($\epsilon=0.2$), the maximum allowable new probability is 0.096, restricting the upward margin to just 0.016. However, under a KL trust region of $\delta=0.05$, Band precisely solves the one-dimensional root (Theorem 1) to reveal a maximum feasible ratio ($r_{\max}$) of roughly 2.41. This means the new probability can safely reach 0.193, allowing a true upward margin of 0.113. PPO artificially chokes this margin at 0.016, whereas the geometric trust region actually permits 0.113. By reallocating the feasible margin according to the trust-region geometry, Band tightens bounds for head tokens and safely expands them for tails. While we do not claim this provides a global sample-complexity guarantee, Band does provide the exact, physically meaningful per-token frontier, thereby eliminating PPO's systematic under-allocation of upward margins.
> > >
> > > **2. Does preserving LP-PA updates improve performance? (Weakness 2)**
> > >
> > > Our goal is not to artificially boost rare tokens, but to fix a structural flaw where PPO unfairly suppresses valid updates *solely* because of their low initial probability. When a tail token has a positive advantage ($A_t > 0$), the unclipped surrogate objective should monotonically increase with the probability ratio $r_t$. However, PPO's fixed threshold forces the absolute upward margin to shrink linearly as $p$ decreases. Consequently, low-probability, positive-advantage (LP-PA) updates hit the ceiling almost immediately, saturating the clipped policy-improvement term and removing any further positive gradient signal from the surrogate.
> > >
> > > Our empirical results demonstrate why fixing this matters. As shown in our isolation experiments (Fig. 4b), fixed-bound methods erroneously truncate 20 to 60 percent of LP-PA updates early in training, while BandPO reduces this inappropriate truncation to near zero. Preserving these valid gradients correlates strongly with the delayed entropy collapse (Fig. 4c) and the consistent performance gains observed across different model scales (Table 1). The improvement stems from fully utilizing the effective policy-improvement gradients available in each batch, rather than artificially cutting them off. BandPO succeeds not simply because it clips less overall, but because it clips strictly where mathematically justified.
> > >
> > > **3. Comparison with entropy regularization (Key Question)**
> > >
> > > Entropy regularization and BandPO tackle exploration from fundamentally different angles. Adding an entropy bonus globally reshapes the optimization objective to forcefully spread probability mass, which requires delicate scheduling of the regularization coefficient to prevent sub-optimal convergence. In contrast, BandPO removes a localized bottleneck in the clipping mechanism without altering the underlying RL objective. It allows natural entropy reduction when the policy confidently identifies optimal reasoning paths, without a generic entropy bonus constantly fighting against convergence. Therefore, the two approaches are orthogonal and could potentially be used in tandem.
> > >
> > > References cited above:
> > > [1] Wang et al., arXiv:2506.01939, 2025.
> > > [2] Huang et al., arXiv:2510.03222, 2025.
> > > [3] Cui et al., arXiv:2505.22617, 2025.

---

### Official Review · Reviewer_Nrnj · 2026-03-17

**Soundness:** 4
**Presentation:** 4
**Significance:** 4
**Originality:** 4
**Overall Recommendation:** 5
**Confidence:** 5

**Summary:**

This paper addresses a fundamental limitation in PPO/GRPO-style clipping mechanisms used for LLM alignment. The authors identify that canonical clipping creates a bottleneck for exploration: when high-advantage actions have low probability under the current policy, the fixed upper clipping bound (either symmetric or asymmetric) severely limits their probability increase, leads to nullifying gradient updates.
The proposed solution, BandPO, replaces fixed clipping thresholds with action-dependent bounds which are derived from f-divergence trust regions. The key insight is that for each action, the 'clipping thresholds' can be obtained by solving a constrained optimization problem that characterizes the maximum allowable divergence from the current policy. Through a clever symmetry argument, the authors reduce this high-dimensional problem (vocabulary size ~50K) to a univariate root-finding task. They provide closed-form solutions for Total Variation and Pearson $\chi^2$ divergences, and propose an efficient bisection method for KL divergence.
The mathematical contribution is significant. Empirically, the authors validate BandPO on autoregressive LLMs across several model scales and demonstrate that BandPO consistently improve over standard clipping on mathematical reasoning benchmarks.

**Compliance With Llm Reviewing Policy:**

Affirmed.

**Final Justification:**

The principled theoretical contribution of projecting f-divergence trust regions into probability-aware per-token clipping bounds is technically sound. The rebuttal adequately addressed my empirical concerns. So, I maintain my score of 5.

**Key Questions For Authors:**

Q1: Comparison with dynamic thresholding methods
Your current baselines only consider fixed thresholds. Can you provide a direct empirical comparison with other dynamic thresholding approaches like DCPO?

Q2: Generalization beyond mathematical reasoning and different reward structures
Since BandPO introduces additional computational complexity, can you provide guidance on when it is best to use it? Specifically, results across different problem types (coding, text generation, logical reasoning) and reward structures (sparse vs. dense) would help understand when the performance gains justify the overhead.

Q3: Computational overhead quantification
For cases requiring numerical solvers (e.g., KL divergence), can you provide timing profiles comparing BandPO against other baselines?Wall-clock time per training step for the exact same setting would clarify the performance-cost tradeoff.

**Limitations:**

Partially. The authors discuss computational overhead and static trust region assumptions in Section 7, but they don't quantify the computational cost or clarify when BandPO is preferable to other dynamic methods like DCPO. The limited experimental scope is also not acknowledged as a limitation.

**Strengths And Weaknesses:**

Soundness:
The theoretical contribution is solid. I worked through the key proofs myself and they are correct. Lemma 1 and Theorem 1 are non-trivial, the reduction from ~50K dimensions to a single-variable optimization is very clever. One minor note: the Lemma 1 proof could be more direct using Lagrangian duality, but their symmetry-based approach is perfectly valid.
However, the empirical evaluation lacks diversity. All experiments focus on mathematical reasoning tasks (AMC, AIME). While math problems have extremely sparse rewards and few correct answers which is exactly the setting where exploration matters most, non-diverse set of experiment limits our understanding of where this method works best. The paper would benefit from experiments on coding tasks, text generation, logical reasoning tasks (e.g., Sudoku), and direct comparisons between sparse vs. dense reward settings to clarify when and why BandPO provides advantages. This is very important given that the KL divergence version requires iterative solvers, which adds computational overhead. So, Understanding the performance-cost tradeoff across different benchmarks would help decide when to use this method.



Presentation:
The paper is well-written and the logical flow from the problem to final solution is clear.




Significance:
The clipping mechanism is fundamental to RL-based fine-tuning of generative models, particularly LLMs. PPO and its variants (GRPO, etc.) are the dominant paradigm for post-training, so improvements to the clipping operator have broad potential impact.



Originality:
The "clipping hurts exploration" problem was identified by prior work. The dynamic bounds solution was explored by DCPO. What's new is the principled derivation, which is valuable.

---

> ### Author Rebuttal · Authors · 2026-03-31
>
> We sincerely thank the reviewer for the careful reading and for checking the key proofs. We are especially encouraged that you recognized the technical value of Lemma 1/Theorem 1 and the broad potential impact of improving clipping-based RL for LLM post-training. Your questions on experimental breadth, computational cost, and practical guidance are highly constructive, and we address them below.
>
> **1. Direct Empirical Comparison with Dynamic Clipping (DCPO) (Key Question 1)**
>
> We fully agree that DCPO is an important dynamic baseline. We implemented DCPO's Dynamic-Adaptive Clipping (DAC) module and conducted a rigorous multi-seed evaluation (3 independent seeds, strictly coupling data and rollout seeds via `verl`) on Qwen2.5-3B-Instruct. Band$_{\mathrm{KL},0.05}$ achieves Avg mean@32 / pass@32 of $22.05\pm0.13$ / $42.49\pm0.71$, versus $21.30\pm0.15$ / $40.88\pm1.44$ for DCPO-DAC and $20.13\pm0.30$ / $39.40\pm1.04$ for Clip-Higher. Beyond higher overall performance, BandPO exhibits notably stronger statistical stability: on the volatile AIME24 pass@32, BandPO yields std $\pm 0.40$ compared to DCPO-DAC's $\pm 3.60$, supporting our claim that mathematically projected bounds are intrinsically more reliable than heuristic rules. Furthermore, extending training to 10 epochs (1,020 steps) revealed that DCPO-DAC suffered catastrophic policy collapse, while BandPO maintained a robust upward trajectory—confirming the structural advantage of $f$-divergence-grounded constraints for long-horizon optimization. The full per-benchmark breakdown is in our response to Reviewer UcPf (Table 1).
>
> **2. Quantification of Computational Overhead (Key Question 3, Limitations)**
>
> We profiled the KL instantiation under the exact same Qwen2.5-3B, 8$\times$H200 setting. Our CUDA-parallelized bisection solver adds merely $\sim$0.67s per actor-update step ($\sim$1.7% of actor-update time), with no measurable memory increase. Crucially, the solver is invoked only during policy-loss computation, never during autoregressive rollout. Since the full RL iteration is dominated by rollout and reward scoring (typically several minutes), this sub-second overhead is virtually negligible end-to-end.
>
> Where minimal micro-operator latency is the sole priority, a heuristic like DCPO-DAC is indeed cheaper ($\sim$0.05s). However, exchanging $\sim$0.6s for mathematically rigorous, probability-aware exploration bounds is a highly favorable tradeoff. Moreover, the TV and Pearson-$\chi^2$ instantiations admit closed-form bounds, sharing the same $O(1)$ per-token complexity as DCPO while retaining the full theoretical framework. The complete profiling is in our response to Reviewer UcPf (Table 2).
>
> **3. Generalization Beyond Mathematical Reasoning and Practical Guidance (Key Question 2)**
>
> Our initial evaluation targeted mathematical reasoning because it represents precisely the regime where the Sec. 4 bottleneck is most severe: sparse/verifiable rewards, long reasoning horizons, and highly concentrated post-SFT policies make positive-advantage tail actions especially susceptible to suppression, since the canonical upward margin scales as $\Delta\pi(a|s)\le\epsilon_+\pi_{\mathrm{old}}(a|s)$. Our training-dynamics analysis (Fig. 4) confirms that BandPO nearly eliminates erroneous clipping of these tokens and preserves entropy far better than fixed clipping.
>
> Regarding practical guidance: the clearest benefit appears in sparse-reward domains (e.g., math reasoning, unit-test-driven code RL) where exploration of rare high-advantage tokens is critical. In denser-reward settings, the exploration bottleneck is less dominant, so BandPO functions primarily as a principled trust-region-aware local controller with potentially smaller performance gaps. Crucially, BandPO is fundamentally task-agnostic—its dynamic bounds derive exclusively from the geometry of the probability simplex ($p$ and $\delta$) via convex optimization, independent of reward structure or domain semantics.
>
> We fully agree that the current empirical scope should be stated more explicitly as a limitation. We will revise the manuscript to include the DCPO comparison and efficiency profiling, add practical guidance on when BandPO is most beneficial, and explicitly position broader task coverage (e.g., code generation, dense-reward settings) as an important next evaluation rather than an already-established conclusion. We thank you again for feedback that significantly strengthens the practical positioning of our work.

---

> > ### Author Rebuttal · Reviewer_Nrnj · 2026-04-03
> >
> > Thank you for the thorough rebuttal. My key questions are addressed and I am satisfied. My remaining concern is the limited experimental scope; I would strongly encourage adding at least one dataset with different characteristics in the camera-ready version. Given the theoretical rigor and negligible computational overhead, I am maintaining my score of 5.

---

> > > ### Author Response · Authors · 2026-04-06
> > >
> > > We sincerely thank you for the careful review and for verifying the key proofs. We are delighted that our responses addressed your concerns about DCPO comparison and computational overhead. Regarding the experimental scope, we are committed to expanding our evaluation to include tasks with different characteristics in the final revision.

---

### Decision · Program_Chairs · 2026-04-30

**Decision:**

Accept (regular)

**Comment:**

**Summary:** This paper studies a limitation of PPO/GRPO-style clipping in LLM reinforcement learning, namely that fixed clipping bounds can disproportionately suppress low-probability but high-advantage actions and thereby accelerate entropy collapse. To address this, the paper proposes BandPO, which replaces fixed clipping with probability-aware per-token bounds obtained by projecting an (f)-divergence trust region onto feasible ratio intervals. The core technical contribution is a reduction of the bound-computation problem to a one-dimensional optimization, with closed-form solutions for some divergences and an efficient numerical procedure for KL. Empirically, the paper shows improved reasoning performance and more favorable training dynamics than standard clipping-based baselines.

**Meta Review:** The paper has several clear strengths. Reviewer Nrnj viewed the theoretical contribution as significant and technically sound, specifically highlighting the reduction from a high-dimensional trust-region problem to a tractable scalar problem. Reviewer 8fXP likewise found the geometric formulation principled and compelling, and after rebuttal considered the added DCPO comparison, efficiency analysis, and clarification of the KL-direction issue sufficient to strengthen the paper substantially. Reviewer vVm5 also positively assessed the core problem insight and theoretical organization, noting that the method preserves the practical convenience of clipping-based RL while replacing heuristic fixed bounds with a more principled alternative. Overall, the review consensus is that the paper makes a meaningful contribution to a central component of RL-based LLM post-training.

There are still a few points that should be addressed carefully in the final version. First, the rebuttal-added results and clarifications should be incorporated into the paper itself, rather than left only in the discussion. In particular, the final version should include the direct comparison with DCPO, the multi-seed validation, the wall-clock and efficiency profiling, and the clarification of the (\delta) selection protocol and its cross-scale sensitivity. Second, while the empirical results are strong, the final paper should present the scope more carefully: several reviewers noted that the evaluation remains concentrated on mathematical reasoning tasks, and the authors themselves acknowledged that broader task coverage, such as code generation or denser-reward settings, should be stated more explicitly as a limitation and as an important direction for future evaluation.

Overall, I recommend acceptance.